

# On biases in atmospheric CO inversions assimilating MOPITT satellite retrievals

**Yi Yin[1],[*],[†], Frederic Chevallier[1], Philippe Ciais[1], Gregoire Broquet[1], Anne Cozic[1], Sophie Szopa[1], Yilong Wang[1]**

[1] Laboratoire des Sciences du Climat et de l'Environnement, CEA-CNRS-UVSQ, UMR8212, Gif-sur-Yvette, France.
[*] Corresponding to: Yi Yin (yi.yin@lsce.ipsl.fr)
[†] Now at: Jet Propulsion Laboratory, California Institute of Technology, Pasadena, CA 91109, USA

## Abstract

Carbon monoxide (CO) inverse modelling studies have so far reported significant discrepancies between model concentrations optimized with the Measurement of Pollution in the Troposphere (MOPITT) satellite retrievals and surface in-situ measurements. Here, we assess how well a global chemistry-transport model (CTM) fits a large variety of independent CO observations (surface and aircraft air sample measurements and ground-based column retrievals) before and after assimilating MOPITTv6 total column ($X_{CO}$) retrievals to optimize CO sources/sinks. Consistent negative prior biases to all types of observations in all sensitivity tests suggest an underestimation of current surface emissions in the Northern hemisphere. In contrast, prior simulations fit the surface air sample observations well in the Southern hemisphere but underestimate CO in the free troposphere and on average in the column. Positive biases in MOPITT retrievals are identified in the Northern mid- and high latitudes, highlighting the importance of proper bias-correction of those satellite retrievals. Biases in representing vertical CO profiles are found over the ocean and most significantly in the Southern hemisphere, suggesting errors in the vertical distribution of CO chemical sources/sinks or in the vertical mixing to be improved in future modelling studies. Varying model-data differences are found in the vertical between CTM and MOPITT retrieved vertical profiles after having assimilated MOPITT $X_{CO}$; these bias structures indicate that the posterior model differences to in-situ observations would be even larger if the near surface retrievals were assimilated instead of $X_{CO}$. In addition, given the higher long-term stability of the MOPITT $X_{CO}$ retrievals as opposed to divergent temporal bias drifts found in the vertical profile retrievals and a lower sensitivity to model errors in the total column quantity than at a certain altitude, we recommend assimilating the column rather than the profiles or the partial profiles at the current stage.



# 1 Introduction

Carbon monoxide (CO) is a pollutant, an ozone precursor, and a key driver of the oxidation capacity of the atmosphere. Its direct natural and anthropogenic sources at
the Earth's surface are mainly from incomplete combustion processes. It is also produced in the atmosphere by the photochemical oxidation of methane ($CH_4$) and non-methane volatile organic compounds (NMVOCs) (Shindell *et al.*, 2006). Being the major sink of hydroxyl radical (OH) in the troposphere, CO influences the tropospheric oxidation capacity and thus the lifetime of $CH_4$, NMVOCs, and other
trace gases whose major sink path is OH (Logan *et al.*, 1981). With an average lifetime of about 1-2 months depending on latitude, CO also serves as a useful proxy tracer for emission and transport of other species that are co-emitted with CO such as $CO_2$ from fossil fuel (Worden *et al.*, 2012) and biomass burning (Yin *et al.*, 2016).

Related to its importance for air quality and climate, CO is one of the best-observed trace gases with air sample measurements from surface stations (Larssen *et al.*, 1999; Novelli *et al.*, 2003), aircraft (Wofsy *et al.*, 2011; Zbinden *et al.*, 2013), and with ground-based total column retrievals from stations in the Total Carbon Column Observing Network (TCCON) (Wunch *et al.*, 2011). CO has also been observed
globally by satellite from several space-borne instruments, like the Measurement of Pollution in the Troposphere (MOPITT) (Deeter *et al.*, 2014) or the Infrared Atmospheric Sounding Interferometer (IASI) (Clerbaux *et al.*, 2009), with some coarse vertical profiling capability for some of the products. The official MOPITT CO products, now reaching version 7, has played a pivotal role in the study of the spatial-
temporal distribution and variability of CO sources and sinks at a global scale (Arellano *et al.*, 2004; Duncan *et al.*, 2007; Fortems-Cheiney *et al.*, 2011; Gaubert *et al.*, 2016; Hooghiemstra *et al.*, 2012; Kopacz *et al.*, 2010; Shindell *et al.*, 2006; Stein *et al.*, 2014; Worden *et al.*, 2013; Yin *et al.*, 2015).

The chemical transport models (CTMs) that link the source/sink processes to atmospheric mixing ratios are key to the interpretation of the CO measurements, particularly in the global atmospheric inversion approach that infers the surface emissions from observed atmospheric mole fraction gradients. Although CTM performances have improved significantly over recent decades, model results are still
plagued by systematic uncertainties due to a complex interplay of source distributions, transport and chemistry, with most CTMs showing negative biases to surface and satellite observations in the Northern hemisphere when prescribed with current emission inventories (Naik *et al.*, 2013; Patra *et al.*, 2011; Shindell *et al.*, 2006).

Satellite retrievals are partial or total column-integrated concentrations, with different vertical sensitivities at different altitudes depending on the instruments. Assimilating surface in-situ measurements or MOPITT CO retrievals using the same CTM and inversion configuration could result in considerable differences in the posterior surface fluxes (Hooghiemstra *et al.*, 2012). MOPITT-based atmospheric
inversions were also shown to be biased high when compared to independent in-situ surface observations in the boundary layer (Gaubert *et al.*, 2016; Yin *et al.*, 2015). Assimilating MOPITT CO profile retrievals for different pressure levels resulted in significant differences in the posterior CO budgets, suggesting some inconsistency in the information provided to the inversion systems on vertical CO distributions



between the model and the observation (Jiang *et al.*, 2015).

In order to diagnose these inconsistencies that could result from model errors or satellite retrieval biases or both, this study evaluates the results of a global MOPITT CO total column assimilation using LMDz-SACS (as described by Yin *et al.* 2015)
against various independent measurements for the period 2009-2011. Three categories of CO measurements are used for the evaluation: i) independent ground-based TCCON total column retrievals, ii) independent in-situ observations from surface networks and aircraft, and iii) MOPITT vertical profile retrievals at ten pressure levels. The observations used here have diverse spatial footprints on surface CO sources and
different sensitivities to horizontal and vertical transport and chemical processes. We also include a series of sensitivity tests (described in Section 2.2 and 2.3) to discuss model uncertainties.

This paper is organized as follows. Section 2 describes the datasets and methods
used in this study, covering successively (i) the MOPITT satellite retrievals, (ii) our CTM, LMDz-SACS, (iii) the prior/posterior simulations and sensitivity tests, and (iv) observations used for model evaluation. In section 3, we first evaluate the model with MOPITT $X_{CO}$ that is used for assimilation, then we present the results of evaluation against independent CO observations from TCCON total column retrievals, surface
station measurements, and aircraft vertical profile measurements. Section 4 represents the assessment against MOPITT vertical profile retrievals before and after assimilating the MOPITT total column retrievals. In section 5, we summarize all evaluation results and discuss potential biases in the model and in the satellite retrieval, as well as their implications for CO inverse study.

## 2    Data and Methods

### 2.1    MOPITT satellite retrievals of CO total column and vertical profiles

The MOPITT instrument flies in a sun-synchronous polar orbit at an altitude of 705 km, onboard NASA's Terra spacecraft launched in December 1999 (Deeter, 2003). It has been providing the longest continuous temporal coverage of global CO measurements from space with a spatial horizontal resolution of $22 \times 22$ km$^2$ since 2000 (with some interruptions in 2001 and 2009). "Multispectral" retrieval products
since version 5 combined near infrared (NIR, around 2.3 µm) and thermal infrared (TIR, around 4.7 µm) radiances and thus offer higher vertical resolution and higher sensitivity to the lower troposphere that are closer to the surface sources (Deeter *et al.*, 2013). In this study, we refer to MOPITT TIR/NIR CO retrieval version 6. It benefits from various improvements given some issues identified in version 5, including a
correction of geolocation bias due to instrument viewing angle, an updated *a priori* CO profile, a higher quality meteorological field for water vapour and temperature, and a radiance bias correction to reduce the latitude-dependent upper troposphere retrieval bias found in V5 (Deeter *et al.*, 2014).

The MOPITT-retrieved total column integrated dry air CO molecule number, noted as $X_{CO}$ hereafter, are calculated in a Bayesian way as

$$C_{rtv} = C_{prior} + \mathbf{A}'(\chi_{mod} - \chi_{prior}) \quad (1)$$





where $C_{prior}$ and $C_{rtv}$ represent the prior and the satellite retrieved total column, expressed as molecule/cm$^2$; $\chi_{prior}$ and $\chi_{mod}$ respectively represent the prior and the modelled vertical CO profiles for the satellite retrieval (here, with $\chi_{mod}$ based on information gain from radiance transfer model). The vertical profiles are resolved at 10 pressure levels, including surface pressure and successively from 900 hPa to 100 hPa with 100 hPa intervals. The vector quantities $\chi$ are expressed as $\log_{10}$ of the corresponding volume mixing ratio. $A'$ is the retrieval averaging kernel matrix, which quantifies the sensitivity of both the instrument and the retrieval to the abundance of CO at different pressure levels (Deeter $et$ $al$., 2013).

The MOPITT CO volume mixing ratios are also retrieved at the 10 pressure levels for the satellite products, expressed as

$$\chi_{rtv} = \chi_{prior} + A(\chi_{mod} - \chi_{prior}) \qquad (2)$$

where $\chi_{prior}$ and $\chi_{mod}$ represent the same quantities as in Equation (1), and $\chi_{rtv}$ represent the retrieved CO profile at the ten pressure levels. The method for coverting $A$ and $A'$ is described in the MOPITT Version 4 User Guide (Deeter, 2009).

We use MOPITT $X_{CO}$ for the global inversion. To calculate the model equivalent $X_{CO}$ to MOPITT, we use Equation (1) with $\chi_{mod}$ being CTM concentrations interpolated at the ten MOPITT pressure levels, together with associated prior CO profiles and averaging kernels. MOPITT retrievals with a solar zenith angle larger than 70°, with latitudes within 25° from the poles, or with surface pressures less than 900 hPa are excluded from the inversion as detailed in Yin $et$ $al$., (2015). The MOPITT vertical profile retrievals are used for model evaluation as described later in Section 2.4.4.

## 2.2 Atmospheric Chemical Transport Model

The transport model used in this study is the global general circulation model LMDz (Hourdin $et$ $al$., 2006), which is part of the Earth system model of the Institut Pierre-Simon Laplace (IPSL). It is coupled with a simplified chemistry module SACS (Pison $et$ $al$., 2009), derived from the full chemistry model – Interaction with Chemistry and Aerosols (INCA) (Hauglustaine, 2004). SACS represents the main species in the chemical oxidation chain of CH$_4$–CH$_2$O–CO; it also represents methyl chloroform (MCF), with OH being the common sink path to relate all the tracers (Yin $et$ $al$., 2015).

The reference version of LMDZ-SACS, used to perform the inversion (briefly described below in Section 2.3), has a vertical resolution of 19 eta-pressure levels and a horizontal resolution of 2.5° (latitude) × 3.75° (longitude), noted as medium resolution (**MR**) hereafter.

We also include models with higher resolution for sensitivity tests, noted as **HR** (High Resolution). They are associated with a higher vertical resolution of 39 eta-pressure levels and a finer horizontal resolution of 1.89° (latitude) × 3.75° (longitude). Version **HR1** has the same physics configuration as MR, using the deep convection scheme of (Tiedtke, 1989) and the boundary layer mixing scheme of (Louis, 1979). **HR2** is an updated version that uses the deep convection scheme of (Emanuel, 1991) and the boundary layer mixing parameterization from (Deardorff, 1966). HR2 corresponds to the version called standard physics (SP) in (Locatelli $et$ $al$., 2015),



where more detail regarding these configurations are described. HR2 has been shown to represent better the deep convection and inter-hemisphere exchange than HR1 as suggested by Radon ($^{222}$Rn) and sulfur hexafluoride (SF$_6$) observations (Locatelli *et al.*, 2015).

For all versions of LMDz-SACS here, sea surface temperature and sea ice coverage from ERA-Interim climate reanalysis produced by the European Centre for Medium-Range Weather Forecasts (ECMWF) are used as boundary conditions; horizontal winds are nudged towards the ERA-Interim wind fields with a relaxing time of 3 hours.

### 2.3    Simulation configurations

In this study, we analyze two sets of global CO simulations for the period from 2009 to 2011: forward simulations prescribed with prior CO sources/sinks and posterior simulations after assimilating MOPITT X$_{CO}$. All simulations are listed in **Table 1**.

#### 2.3.1    Prior forward simulations

All the three LMDz-SACS versions described in Section 2.2 are used to perform the prior forward simulations, noted as **F**. The prior CO sources include: i) bottom-up emission inventories of CH$_4$, CH$_2$O, and CO, and ii) 3D chemical production fields of CH$_2$O from the oxidation of NMVOCs emissions simulated by INCA. The surface emission maps are aggregated from their original inventories to corresponding model resolution (details available in Yin *et al.*, 2015). The 3D chemical CH$_2$O productions are modelled with MR and HR versions of the LMDZ-INCA full chemistry model respectively, given the same emissions and climate forcing (updates from Folberth *et al.*, 2006). The spatial distributions of CH$_2$O production between MR and HR results are similar. One scale factor is applied to the HR results over the globe to conserve the global mass budget to be consistent with the reference version MR.

For CO sink, two different OH fields are used: INCA-OH and TransCom-OH. INCA-OH is produced by the LMDZ-INCA full chemistry model in the same simulation as for chemical CH$_2$O production using MR and HR respectively (updates from Folberth *et al.*, 2006). TransCom-OH was originally prepared for the international TransCom-CH4 experiment of Patra *et al.*, (2011). The TransCom-OH fields are associated with a horizontal resolution of 1 degree and a vertical resolution of 60 levels; they are linearly interpolated to the model resolution of MR and HR respectively. INCA-OH concentrations are generally higher over the Northern Hemisphere (NH) but lower in the high latitudes of Southern Hemisphere (SH) compared to TransCom-OH. Thus, the two OH have very different North-to-South inter-hemisphere ratios, with ~1.2 for INCA and ~1.0 for TransCom. The vertical structures between the two prior OH fields are also different: INCA-OH has higher OH concentrations in the lower troposphere below 700 hPa over the tropics and sub-tropics, but much lower concentrations above the tropopause compared to TransCom-OH (for more information see Figure 3 in Yin *et al.*, (2015)).

#### 2.3.2    Posterior simulations

The posterior CO sources/sinks are optimized using the MR version assimilating





235 simultaneously the MOPITT $X_{CO}$ and surface observations of $CH_4$ and MCF as described in Yin *et al.*, (2015). The HR versions are used for evaluating the posterior fluxes obtained with MR. The posterior simulations are noted as **P**. Two MR inversions associated with INCA and TrasCom OH are performed respectively, as OH fields were shown to induce the largest uncertainty in our prior forward simulations and in previous modelling studies.

240

For the MR inversions, for each weekly window, the surface emission maps of $CH_4$, $CH_2O$, and CO are optimized for each model grid point, the 3D secondary $CH_2O$ sources are optimized by a scaling factor for the entire column at each model grid point, and the 3D OH fields are adjusted by scaling factors over six large latitude 245 bands. That is to say, there are no direct modifications to the vertical distribution of the chemical CO sources in the atmosphere or to the OH fields in the inversion, and the horizontal OH gradients within a large latitude band are not adjusted. More detail regarding the inversion configurations are given in Yin *et al.*, (2015). The MR-optimized surface emissions and scaling factors for $CH_2O$ productions and OH fields 250 are then spatially interpolated to the HR resolution, with global budgets being conserved.

In summary, there are four simulations for each model version (MR/HR1/HR2): prior and posterior simulations associated with INCA or TransCom OH respectively. 255 It is emphasized that posterior HR simulations are prescribed with optimized surface emissions and scaling factors obtained by the MR model with the same OH type. The vertical distribution of $CH_2O$ and OH are not exactly the same due to changes in model resolution. The posterior model performances of HR1 and HR2 are not a direct evaluation of the model ability to fit independent observations, but rather, they 260 demonstrate the model sensitivity due to resolution/parameterization.

### 2.4 Observations for evaluation and statistical analysis

#### 2.4.1 Ground-based CO total column measurements

We use ground-based $X_{CO}$ retrievals from the TCCON 265 (http://www.tccon.caltech.edu/) (Wunch *et al.*, 2011), a network of Fourier transform spectrometers (FTSs) from near-infrared (NIR) solar absorption spectra, designed to retrieve precise total column abundances of $CO_2$, $CH_4$, $N_2O$ and CO. It is noted that the MOPITT NIR/TIR retrievals, combining information from both TIR and NIR, have generally higher sensitivity to the lower troposphere compared to TCCON. 270 Besides, their footprints are also different. The TCCON stations are listed in **Table 2**, with their location indicated in **Figure 1**. The TCCON prior profile and averaging kernel associated with each observation are used to calculate corresponding model retrievals, in a similar Bayesian way as the MOPITT $X_{CO}$.

#### 2.4.2 Surface station measurements

In-situ CO measurements are accessed from the World Data Centre for Greenhouse Gases (WDCGG) database (http://ds.data.jma.go.jp/gmd/wdcgg/, accessed on Dec. 2015). We include CO surface measurements from 88 stations to evaluate modelling results within the boundary layer, calibrated to WMO scale. The 280 geographical distributions of the stations are shown in **Figure 1**, and a list of stations is available in **Table S1**.



### 2.4.3 Aircraft measurements

We use aircraft in-situ CO measurements of vertical profiles from two projects:

1) MOZAIC/IAGOS (Measurements of Ozone and water vapour by in-service AIrbus aircraft / In-service Aircraft for a Global Observing System) (http://www.iagos.org), which contain 20 years (1994 to 2015) of in-situ CO, $O_3$ and NOx measurements, sampled from multiple commercial flight during take-off (ascending) and landing (descending) to achieve quasi-vertical profiles (Marenco *et al.*, 1998; Petetin *et al.*, 2015). The observations are thus close to international airports and hence represent an environment of high anthropogenic emissions, in particular in the layers close to the surface.

2) High-Performance Instrumented Airborne Platform for Environmental Research (HIAPER) Pole-to-Pole Observations (HIPPO) large-scale aircraft campaigns, which sampled the atmosphere from the North Pole to the coastal waters of Antarctica over the Pacific Ocean (https://www.eol.ucar.edu/field_projects/hippo) (Wofsy and HIPPO Science Team and Cooperating Modellers and Satellite Teams, 2011). The flights took air samples from the surface up to 14 km on a campaign basis, spanning different seasons in different years (I: Jan. 2009, II: Nov. 2009, III: Mar.-Apr. 2010, IV: Jun.-Jul. 2011, V: Aug. – Sep. 2011).

Individual aircraft profiles are assigned to certain model grid points given their geographic location and pressure levels, and all measurements are then averaged per model grid point at a 30-minute resolution to compare with corresponding model value. Only observations over land are kept for the MOZAIC/IAGOS measurements for further analysis; while only observations over the ocean are kept for the HIPPO measurements. Their spatial coverages are shown at the model resolution in **Figure 1**.

### 2.4.4 MOPITT CO vertical profile retrievals

The MOPITT vertical profile retrievals are used for evaluation. As the MOPITT total columns are assimilated, the profiles are not independent of the posterior simulations. The analysis of posterior models would thus show how well the model agrees with the MOPITT profile when fitting the $X_{CO}$. The MOPITT-equivalent model CO vertical profiles are retrieved in a consistent manner as the satellite retrievals using Equation (2) described in section 2.1, in which the $\chi_{mod}$ become the CTM concentrations interpolated at the then MOPITT pressure levels to be convolved with corresponding MOPITT prior CO profiles and averaging kernels.

### 2.4.5 Model sampling and statistical analysis

Model values are sampled according to the time and location of the measurements at a temporal resolution of 30 minutes and the model associated spatial resolution. In the case of surface and aircraft measurements, no horizontal or vertical interpolation among nearby model grid points is made to smooth the sampling. In the case of interpolating the model vertical profile to other instrumental levels to obtain $\chi_{mod}$ at the instrument vertical resolution, e.g. from the model 39 vertical levels to the 10 MOPITT pressure levels, the total column of the model state integral is always conserved to compensate uncertainty from vertical resolution change on the CTM side.





The same applies for TCCON $X_{CO}$ retrievals as well. For each dataset, the observed and simulated values are averaged per month for further analysis in this study, e.g. the model-data biases are calculated using monthly averages of model-equivalences minus corresponding observations.

## 335   3   Results

### 3.1   Evaluation against the MOPITT total column ($X_{CO}$)

The latitudinal mean bias of simulated CO compared to MOPITT $X_{CO}$ is shown in **Figure 2a-b**. The prior forward model results show some significant
underestimation of CO concentrations compared to the MOPITT $X_{CO}$, particularly in the northern hemisphere, irrespective of the model version or of the OH field (shown by dotted lines). Logically, the posterior model results agree with MOPITT $X_{CO}$ much better than the forward run before optimization, with a mean bias of less than 0.5 ppb for both OH in the reference MR version (shown by solid red lines).  Small positive
biases using the HR models are noticed (shown by solid green/blue lines). The statistics of model biases for four latitudinal bands are summarized in **Table 3**.

For the reference transport model version MR, the mean $X_{CO}$ bias of the prior forward modelling (shown by dotted-red lines in **Figure 2a-b**) is -12.1±7.6 ppb and -
9.4±3.7 ppb for INCA-OH and TransCom-OH respectively (with the range showing 1-sigma standard deviation of the mean biases of all model grids, the same applies hereafter if not specified otherwise). INCA-OH produces a larger negative bias to MOPITT in the Northern high latitudes than TransCom-OH, because of its higher OH concentration in the NH. Accordingly, the inversion using INCA-OH gives ~17.9%
higher NH emissions, but ~18.4% lower SH emissions than the inversion using TransCom-OH, resulting into ~6.1% higher global emissions in the INCA-OH based inversion. It is noted that although we optimize OH together with surface emissions, the system only scales slightly the big-region OH state vectors (in total six big regions over the globe) and thus the inverted surface emissions are sensitive to the prior OH
fields.

The two HR models (shown in green and blue in **Figure 2a-b**) produce slightly larger $X_{CO}$ values than the MR model when being forced by either prior or optimized CO fluxes. The CO surface sources are conserved in global mass between different
resolutions when emitted to the atmosphere, but with the change in resolution, the CO sink via the reaction with OH (associated with different resolutions) may differ. For the global average, the HR models generate ~1.0 ppb (~1.5%) and ~2.0 ppb (~3%) higher $X_{CO}$ compared to the MR, using INCA-OH and TransCom-OH respectively.

The difference between the HR and MR $X_{CO}$ results (~1.5 ppb) is of a much smaller magnitude than the differences between the prior and the posterior MR simulations (~10.4 ppb); it is also of a smaller magnitude than that induced by the two OH fields in the prior forward simulations when the CO sources are identical (~2.8 ppb for global average with some cancelling effect between the NH and SH). The
differences in modelled $X_{CO}$ between HR1 and HR2 are not significant at a global scale.



### 3.2 Evaluation against TCCON column measurements

The average biases of modelled $X_{CO}$ compared to TCCON measurements along latitude are shown in **Figure 2c-d**. The prior forward simulations show general underestimation of $X_{CO}$ compared to TCCON measurements (shown in light colours in **Figure 2c-d**), consistent with the negative bias against MOPITT $X_{CO}$ (dotted lines in **Figure 2a-b**). This negative bias is of -16.0±6.0 ppb with INCA-OH and -9.3± 4.0 ppb with TransCom-OH on global average. The posterior MR results (shown in red dots in **Figure 2c-d**), however, overestimate $X_{CO}$ in the northern mid and high latitudes compared to TCCON, resulting in a positive bias of 7.4±6.4 ppb on average in NH using INCA-OH (7.3±6.0 ppb using TransCom-OH). They also slightly overestimate $X_{CO}$ in SH, resulting in a smaller positive bias 2.5±3.1 ppb using INCA-OH (3.0±2.2 ppb using TransCom-OH).

The average prior forward model difference in TCCON equivalent $X_{CO}$ between the two OH is ~7 ppb, inducing the largest uncertainty among factors tested here (difference between **Figure 2c** and **2d**). The HR models (shown in green and blue) produce ~2.8 ppb (~3.6%) higher concentrations than the MR (shown in red), when using the same type of CO sources and OH profiles (but different in resolution). The relative differences of the column mixing ratios between HR1 and HR2 are not significant, suggesting that modelling the column is not very sensitive to the choice of deep convection and vertical diffusion schemes, however the model resolution does have a visible impact (HR vs. MR). Statistical analysis of model biases and model-data correlations for individual stations are shown in **Table S2**.

### 3.3 Evaluation against independent surface in-situ measurements

The average biases of model CO dry air mole fraction [CO] compared to surface in-situ measurements along latitudinal gradient are shown in **Figure 2e-f**. In the prior forward simulations (shown in light colours), the reference model (MR) generally agrees well with the surface observations in the SH, with only a small negative bias in the SH sub-tropics. The mean MR prior biases in SH are -2.5±6.5 ppb and -4.6±6.8 ppb with INCA and TransCom OH respectively, thus different from the significant negative prior bias compared to MOPITT and TCCON $X_{CO}$. However, negative prior biases increase with latitude in the NH, peak around 50°N, and remain relatively high in the NH high latitudes. The mean MR prior biases in NH are -30.6±12.6 ppb and -20.1±12.7 ppb with INCA and TransCom OH respectively (**Figure 2e-f; Table 2**).

The posterior MR models slightly overestimate in-situ [CO] in the SH mid-to-high latitudes, with a change of sign and a degradation of the bias to surface station data compared to the prior simulations (mean posterior bias in SH: 7.5±8.1 ppb with INCA-OH and 8.2±8.3 ppb with TransCom-OH). The posterior modelled [CO] agrees quite well with ground observations in the NH sub-tropics, but show significant overestimation North of 30°N, i.e. a change in the sign and a slightly smaller amplitude of bias (mean bias in NH: 7.6±39.2 ppb with INCA-OH and 18.2±23.8 ppb with TransCom-OH), as compared to the forward simulations (**Figure 2e-f**). The average difference between the HR and MR model is 3.9±4.1 ppb (~4%) when other setups being the same; here, the 1σ standard deviation showing the spread across prior/posterior simulations and between the two HR models. HR2 simulates slightly





425 higher surface CO concentrations than HR1 (1.4±0.6 ppb, ~1%). Larger differences of model results among different model configurations are found in the surface [CO] than in the quantity of the column integral, suggesting a higher sensitivity in modelling boundary layer mixing ratios.

430 In the prior forward simulations, significant correlations between the model-data difference in surface [CO] and in MOPITT $X_{CO}$ at the same horizontal model grid are found for all three models in the NH (r=0.72 for INCA-OH and r=0.63 for TransCom-OH on average, **Figure 3a**). This result suggests that both surface and satellite observations give consistent information on the errors of the prior CO sources/sinks in 435 the NH, regardless of the model version or OH type. No such significant correlations are found in the posterior simulations over the NH (**Figure 3b**), suggesting no consistent systematic errors of the MOPITT $X_{CO}$ assimilation compared to surface [CO] measurements. On the contrary, in the SH prior forward simulations, no significant correlations between the model-data difference in surface [CO] and in 440 MOPITT $X_{CO}$ are found (**Figure 3c**), suggesting different patterns of the prior model-data deviations when compared to in-situ observations in the boundary layer and to the satellite $X_{CO}$ retrievals. However, a negative correlation (r=-0.49) is noticed for the posterior simulations using HR version (not for the MR version) associated with INCA-OH, suggesting contradicting model-data agreements at the surface and in the 445 column likely due to error in the vertical CO distribution (**Figure 3d**).

### 3.4 Evaluation against independent aircraft measurements

#### 3.4.1 MOZAIC measurements over large airports

MOZAIC aircraft profiles and corresponding model values are shown for four 450 latitudinal bands on the left panel of **Figure 4** (results with INCA-OH are shown). Globally averaged model biases to MOZAIC measurements and variations of model-data correlations in the vertical profiles are shown on the left panel of **Figure 5** (results with both OH fields). The forward model profiles show negative biases regardless of model version or OH fields (shown in dotted lines). The negative biases 455 are larger in the NH than in the SH, consistent with the different in prior model column-integrated $X_{CO}$ against MOPITT and TCCON.

The negative model-data biases in the prior model are well corrected in the posterior simulations (presented by solid lines). The zonal vertical gradients are 460 generally well captured by the model (**Figure 4**). Compared to MOZAIC profiles, the HR models represent the vertical gradients and boundary layer better than the MR with a two-fold higher vertical resolution, i.e. the HR versions produce sharper vertical gradients from the surface up to around 800 hPa, in better agreement with the profile observation.


Apart from the general good representation of the MOZAIC profiles, modelled CO concentrations are considerably lower in the near ground layers compared to the MOZAIC measurements. This could be partly explained by non-modelled local contamination of MOZAIC data in the boundary layer over large international airports, 470 given the limited resolution of our global model to resolve city scale emissions. Another model deficiency compared to MOZAIC profiles is a positive bias in the upper troposphere above 400 hPa (**Figure 5**), which becomes larger to the upper levels, mostly contributed by the NH, in particular, the high latitudes (**Figure 4**).




Along with the increase of model biases in the vertical, model-data correlations also degrade significantly above 500 hPa, with the lowest correlations at around 200 hPa. HR versions show larger decreases in the model-data correlations in the upper troposphere compared to MR. The vertical distributions of model-data correlations between the two OH fields are similar, but INCA-OH simulates slightly smaller positive biases in the upper troposphere than TransCom-OH (**Figure 5**).


### 3.4.2    HIPPO measurements over the Pacific Ocean

The HIPPO aircraft profiles and corresponding model values (with INCA-OH) are shown for four latitudinal bands on the right panel of **Figure 4**. Similar to that of the MOZAIC measurements, the global average biases to HIPPO profiles and vertical
variations of model-data correlations are shown on the right panel of **Figure 5**. The forward models (shown in dotted lines) significantly underestimate the CO concentrations nearly across the entire vertical profile in the NH against HIPPO. However, only small negative prior biases, limited to the upper troposphere, are found in the SH (**Figure 4**). This pattern is consistent with the comparison to surface
network measurements and column integrated $X_{CO}$ mixing ratios.

The posterior simulations show improvement of the overall agreement with HIPPO profiles in the NH, but they overestimate the lower troposphere concentrations, which is in agreement with the comparison to surface [CO] measurements. However,
over the SH, the forward models capture the lower tropospheric (from the surface to 800 hPa) CO concentrations better than the posterior models (**Figure 4**), consistent with the small negative prior bias but significant positive posterior biases to surface observations (**Figure 2e**). The posterior models also overestimate the top layers above ~300 hPa, similar to the feature found in comparison with MOZAIC. The model-data
correlations also drop significantly around 200 hPa, with a larger deterioration with TransCom-OH than with INCA-OH (**Figure 5**). This pressure level is around the tropopause depending on the latitude and season. The HR models simulate slightly higher concentrations across the entire profile, consistent with a higher column integrated mixing ratio equivalent to MOPITT or TCCON; they also represent the
boundary layer better.

Note that the HIPPO measurements were made over the Pacific Ocean, where surface CO emissions are small and the CO sources are from secondary chemical sources or transport. This is also partly the reason for the observed increasing
gradients from the surface toward the upper troposphere, up to ~500 hPa in the NH and to ~300 hPa in the SH (**Figure 4**). Such bias in representing the oceanic vertical profiles suggests error in the vertical distribution of CO source/sink over ocean or in the vertical mixing.


## 4    Evaluation against MOPITT vertical profiles

After evaluating the model with independent observations, we now assess the simulations against MOPITT CO vertical profile retrievals. The latitudinal average
MOPITT CO profiles and equivalent model retrievals (MR and HR1 using INCA-OH) are shown in **Figure 6**. Compared to the MOPITT profiles (**Figure 6a**), the forward





simulations (**Figure 6b-c**) significantly underestimate the CO mixing rations across the vertical profiles in the NH mid-high latitudes, consistent with the comparison to TCCON $X_{CO}$ retrievals, and in-situ measurements from surface stations and MOZAIC aircrafts. The posterior vertical model profiles (**Figure 6e-f**) show better agreement with MOPITT profiles than the forward models, but there are still visible differences.

The average zonal profiles of the observation (solid lines), forward (dotted lines) and posterior (dash-dot lines) model results for the reference model (MR) are shown in **Figure 6d**. We focus here on the posterior model results, after assimilation of the MOPITT column $X_{CO}$ thus not independent from the MOPITT profiles. The MOPITT vertical profiles are well reproduced by the model north of 30°N, but considerable deviations are found for the vertical structure over the tropics and subtropics. The vertical gradient of the CO decrease from the surface to the upper atmosphere is larger in the MOPITT profiles than in the MR model over the tropics and subtropics, in particular in the SH tropics. In the SH mid-high latitudes, the model produces lower concentrations at the surface but higher in the upper troposphere compared to MOPITT, contrary to the difference between the model states and HIPPO profiles. Significant CO enhancements around pressure level 200 hPa are noticed in the MOPITT profiles over the Tropics (**Figure 6a**), which persist throughout the year (**Fig. S1**). Such large CO enhancements in the upper troposphere are not represented by any version of the model; they are also not very visible in the HIPPO or MOZAIC profiles (**Figure 4**).

Comparing model results against MOPITT profile data at the surface level (higher than 900 hPa), the absolute biases in the prior simulation are significantly reduced after assimilating MOPITT $X_{CO}$. The prior surface layer zonal biases (MR) are -23.0 ppb, -4.4 ppb, -1.5 ppb, and -8.1 ppb for the four latitudinal bands from the north to the south, whereas the posterior biases (MR) are reduced to -6.5 ppb, +2.5 ppb, 1.6 ppb, and -5.4 ppb, with the signs changed over the tropics (**Figure 7**). The zonal average posterior model difference to MOPITT profiles remain negative for the mid-high latitudes in both NH and SH from the surface to ~700 hPa; On the contrary, the posterior model difference to MOPITT profiles become positive in the tropics (**Figure 7**).

For regional detail, the relative bias and the model-data correlation with MOPITT profiles for each pressure level are shown as pseudo Taylor diagrams in **Figure 8**. For all regions over land, the posterior simulations (shown by filled markers) agree better with MOPITT profiles than the prior (shown by empty markers). The posterior model-data correlation coefficients are higher than 0.9 for all regions for both $X_{CO}$ and the near-surface level of the profiles. The model-data correlations decrease significantly with increasing altitude, suggesting that the model ability to represent the spatial distribution of MOPITT-based CO concentrations degrades towards the upper atmosphere. The poorest model-data correlations are found around 300 hPa, except for 200 hPa over the tropical ocean, in agreement with the comparison to aircraft measurements.

The posterior model CO profiles match the MOPITT profiles reasonably well over North America, Europe, Boreal Eurasia, Middle East, and Australia (**Figure 8**). Significant divergences of the posterior profiles vs. the MOPITT profiles are found over South America, Africa, Temperate Asia, and Indonesia, where the model



overestimates CO concentrations in the lower part of the atmosphere (900-600 hPa) but underestimates CO concentrations in the upper atmosphere. The posterior models also produce some positive biases over ocean, from the surface up to 400 hPa, although the column integrated $X_{CO}$ are well fitted. Such misfit is most prominent over the tropical ocean, which is related to the MOPITT CO enhancement around 200 hPa as mentioned above.

The largest differences due to model configurations are induced by different OH fields, more than by model resolution or the choice of a convection scheme (**Figure 7**). The relative differences between the HR and MR retrieved MOPITT-equivalent CO profiles are not very significant at the regional scale (shown by different markers in **Figure 8**). The HR versions generally produce higher near-surface concentrations and stronger vertical gradients, thus being more in agreement with the structure of the MOPITT profile. However, these differences induced by model configuration are small; all model versions are associated with the vertically varying posterior model difference to the MOPITT profiles when MOPITT $X_{CO}$ are reasonably fitted (**Figure 7**). The relative change of model differences to MOPITT vertical profiles between HR and MR or between the two HR versions are consistent no matter with OH field is used.

# 5    Discussion and conclusions

## 5.1    Sensitivity to model configurations

Using both our prior and MOPITT $X_{CO}$-assimilated CO sources/sinks, we evaluated the distribution of modelled CO mixing ratio in the atmosphere against independent observations from TCCON total column $X_{CO}$, surface measurements in the boundary layer, and aircraft profile measurements using six sets of model configurations. The choice of the prior OH seems to produce the largest differences in the simulated CO concentrations at a global scale. As the two prior OH field tested here have contrasted North-to-South distribution, the inversion using INCA-OH derives ~17.9% higher NH emissions, but ~18.4% lower SH emissions than the inversion using TransCom-OH, result in ~6.1% higher global emissions associated with INCA-OH. Although OH is optimized jointly with the surface emissions and atmospheric productions of CO, the system does not apply a strong correction to the big-region OH state vectors. The posterior OH fields stay very close to the prior ones, with scaling factors ranging from 0.97 to 1.01. Thus, the different N-S gradients between INCA and TransCom OH remain similar in the posterior.

TransCom-OH, with a North-to-South ratio of around 1, produce slightly smaller surface bias over the NH (**Figure 2e-f**) and also a smaller bias in the modelled seasonal cycle amplitude at the surface (**Figure S2**). The North-to-South gradient in TransCom-OH is also closer to a recent observation-derived near equal N/S OH distribution (Patra *et al.*, 2014). However, larger biases above 300 hPa are found with TransCom-OH, likely caused by inconsistencies between the imported climatological TransCom-OH (produced by another model) and the LMDz model physics, e.g. the difference in tropopause height between the model physics and that implied by the TransCom-OH profiles simulated by a different CTM (**Figure 7**). Our results,



therefore, highlight the importance of both horizontal and vertical OH structures in modelling the CO vertical profiles, with upper atmospheric OH concentrations having a larger impact on tracers with longer lifetimes.

Compared to the reference version (MR), the HR versions, with a nearly two-fold increase in the vertical levels and a factor of one-third increase in latitudinal resolution, simulate 2% higher MOPITT-equivalent-$X_{CO}$, 3.6% higher TCCON-equivalent-$X_{CO}$, and 3.9% higher surface CO mixing ratio at sites from the surface network when averaged globally. The differences between the two convection schemes (HR1 vs. HR2) are not significant for the column integrated $X_{CO}$; but for the
boundary layer, HR2 simulates slightly higher mixing ratios, in agreement with previous studies showing that the HR2 scheme represents deep convection more rigorously, agreeing better with $^{222}$Rn observations (Locatelli *et al*., 2015). The differences between the two HR schemes associated with different physics and convection schemes are much smaller than the differences between the HR and the
MR version, illustrating the importance of an appropriate vertical resolution to represent vertical mixing and the boundary layer.

No significant correlations are found between the relative differences in surface [CO] and column $X_{CO}$, for either HR vs. MR or HR2 vs. HR1 (**Figure S3**), suggesting
that the model does not systematically produce higher surface [CO] when it produces higher $X_{CO}$. The small but consistent increase in the modelled $X_{CO}$ from MR to HR indicates that assimilating MOPITT $X_{CO}$ with a higher resolution model is likely to yield a slightly smaller surface emission compared to the MR results.

**5.2    Possible biases in the prior CO sources**

Despite the uncertainty due to model configurations, particularly OH, our analysis shows that the prior simulations appear to underestimate the CO concentrations persistently in the NH at the surface and at all altitudes, thus result in a negative model bias for the total column (**Figure 2** and **Figure 4**). We also find
consistent model-data deviations in the surface and in the MOPITT Xco vary in space (**Figure 3a**), suggesting an underestimation of NH CO emissions in the current inventories, in line with previous model studies. For instance, Stein *et al*., (2014) showed that a higher winter traffic emissions from North America and Europe and a lower dry deposition rate could improve the agreement between simulated and
observed CO during winter and spring.

In the SH, however, prior simulations fit the surface in situ observations well, but show negative biases to the aircraft profiles and $X_{CO}$ (**Figure 2, Figure 3c,** and **Figure 4)**, suggesting model errors in representing the vertical distribution of CO source/sink,
particularly over the ocean. This could be partly caused by errors in the vertical distribution of tropospheric secondary chemical CO sources, as the surface emissions are generally low but important sources are from chemical oxidation of hydrocarbons and from long-distance transport (Zeng *et al*., 2015). From this aspect, assimilating $CH_2O$ would in principle improve the representation of chemical CO sources
(Fortems-Cheiney *et al*., 2012), future studies to improve the vertical distribution of secondary CO sources would improve the modelling of CO profiles.

Another possible source of errors in modelling the SH vertical CO profiles could



be a misrepresentation of the emission injection height of biomass burning by thermal
plumes. In our current model, pyrogenic emissions are released in the surface layer of
the model, in the same way as fossil fuel emissions, and the thermal impact of air
uplift is not particularly considered. Although most of the injection heights (>80%)
were observed to remain within the boundary layer, some occasional injections could
exceed 8 km and may contribute to some of the positive model bias at the surface
(Gonzi *et al*., 2015; Labonne *et al*., 2007), which may cause some errors in
representing the effects of fire emissions on the vertical structure of CO, as well as of
other CO precursors that are co-emitted from fires. For instance, the MOZAIC
measurements (mainly sampled over Africa as shown in **Figure 1**) show much higher
CO concentrations between 900 hPa and 800 hPa than the model in the sub-tropical
land in SH (0-30S) (**Figure 4**), where CO plumes are primarily from biomass burning.

### 5.3    Possible biases in the MOPITT retrievals

Although the prior model biases to the above-mentioned independent
observations are significantly reduced in most cases after assimilating the MOPITT
$X_{CO}$, there are noticeable remaining model biases in the posterior with a change in the
sign from negative in the prior modelling to positive in the posterior. The posterior
simulations show, on the one hand, consistent positive biases to independent
observations including TCCON $X_{CO}$, surface observations, and HIPPO vertical
profiles in the mid- and high latitudes (**Figure 2** and **Figure 4**); on the other hand,
negative biases to MOPITT vertical profile retrievals for the near surface levels (from
surface to 700 hPa) are seen over these regions (**Figure 7**). Such discrepancy indicates
positive biases in the MOPITT retrievals for the total column and at the near surface
levels in the mid- and high latitudes. As for the tropics and subtropics, the CO
enhancements around 200 hPa in the MOPITT vertical profile retrievals are not found
in the HIPPO or MOZAIC aircraft measurements.

A recent study directly compared the MOPITT retrievals against HIPPO aircraft
measurements and suggested positive $X_{CO}$ biases in the mid-latitudes (40-70°N and
30-60 °S) and negative biases in the tropics (**Figure 1** in (Jiang *et al*., 2016)). Their
results also showed a similar latitudinal bias pattern in the retrieved CO mixing ratios
in the lower troposphere as in the $X_{CO}$, associated with an opposite latitudinal bias
pattern in the upper troposphere retrievals. It is however noted that these evaluations,
in agreement with our model-data bias to HIPPO, were based primarily on
observations over the ocean made during reduced periods of time, while the
comparison of posterior models to MOZAIC profiles does not show such a significant
bias in our analysis (**Figure 4**). Therefore, more in-depth analysis is needed to bias-
correct the satellite data well.

Compared to the IASI data (v20100815), MOPITT total columns (V5T) are
generally higher over land (ranging from 0-13 %) for the overlapping period of the
two instruments from 2008 to 2013; Significant differences are due to a higher prior
CO in the MOPITT retrievals, but biases remain when reprocessing the MOPITT data
using the same a priori constraints as those used for IASI (George *et al*., 2015). The
differences between these two instruments are significantly larger than that between
MOPITT v5 and v6 [*Deeter et al.,* 2014]. More study to reconcile recent satellite CO
retrievals that have improved quality and the combined use of multiple instruments as
explored in (Kopacz *et al*., 2010) would benefit our understanding regarding CO



source/sink distributions.

### 5.4 Assimilating satellite total column vs. vertical profile retrievals

Significant divergences of the differences between the model and MOPITT retrieved vertical profiles are found in the vertical among the ten MOPITT product pressure levels (**Figure 7**, **Figure 8**, and **Figure S1**). Such a spread would certainly result in significant differences in the inverted CO budgets when assimilating one retrieval level at a time, i.e. as suggested in (Jiang *et al.*, 2015) , if there is no change in the model configuration that could impact the vertical CO structures (e.g. CO source/sink vertical distribution). With remaining negative posterior model biases to MOPITT profiles in the lower troposphere over the NH and SH mid- and high latitudes where significant positive model bias to surface and TCCON observations are found, we anticipate that assimilating those profile retrievals of near surface levels (from surface up to 700 hPa) would results in even larger positive biases (**Figure 7**). Similar divergences were reported in previous studies using other CTMs to assimilate MOPITT profiles or near-surface levels (Gaubert *et al.*, 2016; Hooghiemstra *et al.*, 2012; Jiang *et al.*, 2015). Jiang *et al.*, (2016) introduced a 4-order polynomial curve to correct the latitudinal dependent bias for the MOPITT profiles at each retrieval pressure level before the assimilation, comparing to HIPPO measurements as described above in section 5.1. This approach could reduce the bias for a certain pressure level, but it may degrade the overall column consistency and is not well constrained.

In terms of long-term stability, the bias drift for the MOPITT V6 TIR/NIR product varies from negative in the lower troposphere ($-1.3\%\text{yr}^{-1}$ at 800 hPa) to positive in the upper troposphere ($1.6\%\text{yr}^{-1}$ at 200 hPa) as compared to aircraft measurements; yet, the bias drift in the total column is negligible (0.003%/year) (Deeter *et al.*, 2014). Therefore, the column measurements are more suitable for studies associated with long-term CO dynamics. In the meantime, the number of retrieved layers exceeds the number of independent pieces of information available vertically and hence the profiles are only weakly resolved, not representative of the vertical resolution of the observation. The error covariance among different layers of the profile retrievals is described in the full averaging kernel matrix, however we do not have information about the profile error covariance of the CTM, l*et al*one the fact that the measurement error correlations are commonly ignored in inversion systems for technical reasons.

From the CTM perspective, the evaluation against aircraft measurements reveals significant model errors in representing the vertical CO gradient, in particular over the ocean (**Figure 4**). Assimilating satellite retrievals at a particular pressure level would be more prone to model biases in the vertical profiles, which are sensitive to the surface pressure, the vertical distribution of chemistry-related source and sink terms, the vertical mixing, and the lateral transport at a given height. Rayner and O'Brien, (2001), Olsen and Randerson (2004), and Yang *et al.* (2007) argued similarly for $CO_2$ that the assimilation of column-integrated data is less susceptible to uncertainty in model transport than the surface network. The argument may be even stronger for CO, because its relatively short lifetime makes the column integrated concentration more informative about regional emissions.



On either the satellite retrieval side or the model side, representing the vertical profile correctly seems to be a grand challenge. Therefore, we argue that for the purpose of top-down estimates of CO emissions, in which the model cannot directly
correct vertical model biases, it is more robust to assimilate the column than a particular pressure level retrieval, a partial profile, or the entire profile without the full AKs at the current stage. In terms of bias-correction, the use of a bias-correction scheme based on differences between satellite retrievals and a CO reanalysis assimilated with surface and aircraft measurements would be an alternative approach.
More in-depth study with independent observations from various platforms, as well as from other satellite instruments, for instance IASI as explored in (George *et al.*, 2015), would be needed.

The model uncertainty that we discuss in this study is limited to the model
representation of vertical profiles. Other aspects like lateral transport or NH-SH air exchange rate are not addressed. The horizontal and vertical distributions of OH induce the largest uncertainty in the modelling of global CO and thus its atmospheric inversion. The vertical distribution of chemical CO sources is the other key factor toward reconciling model agreement with $X_{CO}$ and in-situ air samples. Future model
studies would benefit from the use of multiple tracers, given that more than half of the CO sources are from secondary chemical oxidation of $CH_4$ and NMVOCs in the troposphere, and $O_3$, NOx are chemically related with CO through OH (Miyazaki *et al.*, 2015). The vertical distributions of all the tracers, measured from aircraft or satellite, would provide important information to improve our understating regarding
the CO source/sinks, as well as the uncertainties of transport and chemistry in CTMs.



## Acknowledgements


This study is supported by the European Research Council Synergy grant ERC-2013-SyG 610028 (IMBALANCE-P). Data to support this article can be obtained by contacting the corresponding author. We wish to thank the NCAR MOPITT team to
make the CO retrieval products publicly available through ftp://l5eil01.larc.nasa.gov/MOPITT/MOP02J.006. We thank the TCCON team for maintaining the network; TCCON data were obtained from the Data Archive operated by the California Institute of Technology from the website http://tccon.ipac.caltech.edu/. Support for TCCON is provided by many national
research support organizations that are listed on the TCCON web site. We thank the World Data Centre for Greenhouse Gases (WDCGG) for archiving surface CO measurements from various stations over the globe; the data were obtained from http://ds.data.jma.go.jp/gmd/wdcgg/. We thank the MOZAIC-IAGOS team for maintaining the airline measurements with data available via http://www.iagos.org/.
We thank the entire HIPPO team for making these measurements available via http://hippo.ucar.edu/. The authors are very grateful to the many people involved in the surface and aircraft measurement and in the archiving of these data.



# Tables and Figures

**Table 1. List of simulations in this study.** We include three model versions (MR/HR1/HR2), and for each version there are prior forward and posterior simulations (F/P) associated with two different OH fields INCA or TransCom (in/tc).

| Model version | Physics | Simulation | CO sources | OH |
|---|---|---|---|---|
| **MR\*** (2.5°x3.75°xL19) | Tiedtke (1989) | MR_F_in | Prior | INCA |
| | | MR_F_tc | | TransCom |
| | | MR_P_in | Posterior_in | Posterior_in |
| | | MR_P_tc | Posterior_tc | Posterior_tc |
| **HR1** (1.895°x3.75°xL39) | Tiedtke (1989) | HR1_F_in | Prior | INCA |
| | | HR1_F_tc | | TransCom |
| | | HR1_P_in | Posterior_in | Posterior_in |
| | | HR1_P_tc | Posterior_tc | Posterior_tc |
| **HR2** (1.895°x3.75°xL39) | Emanuel (1991) | HR2_F_in | Prior | INCA |
| | | HR2_F_tc | | TransCom |
| | | HR2_P_in | Posterior_in | Posterior_in |
| | | HR2_P_tc | Posterior_tc | Posterior_tc |

*\* The only model version used for inversion to obtain optimized CO sources and sinks with INCA or TransCom OH respectively.*

**Table 2 TCCON measurement sites used for the evaluation.**

| Station | latitude | longitude | Altitude (m) | Reference |
|---|---|---|---|---|
| Bialystok, Poland | 53.2 | 23.0 | 180 | (Deutscher *et al.*, 2014) |
| Bremen, Germany | 53.1 | 8.9 | 27 | (Notholt *et al.*, 2014) |
| Darwin, Australia | -12.4 | 130.9 | 30 | (Griffith *et al.*, 2014a) |
| Eureka, Canada | 80.1 | -86.4 | 610 | (Strong *et al.*, 2014) |
| Garmisch, Germany | 47.5 | 11.1 | 740 | (Sussmann and Rettinger, 2014) |
| Izana, Tenerife | 28.3 | -16.5 | 2370 | (Blumenstock *et al.*, 2014) |
| JPL, USA | 34.1 | -118.1 | 230 | (Wennberg *et al.*, 2014a) |
| Kalsruhe, Germany | 49.1 | 8.4 | 116 | (Hase *et al.*, 2014) |
| Lamont, USA | 36.6 | -97.5 | 320 | (Wennberg *et al.*, 2014b) |
| Lauder, New Zealand | -45.0 | 169.7 | 370 | (Sherlock *et al.*, 2014a, 2014b) |
| Orleans, France | 48.0 | 2.1 | 130 | (Warneke *et al.*, 2014) |
| Park Falls, USA | 48.5 | 2.4 | 60 | (Wennberg *et al.*, 2014c) |
| Reunion Island | -20.9 | 55.5 | 87 | (De Mazière *et al.*, 2014) |
| Saga, Japan | 33.2 | 130.3 | 7 | (Kawakami *et al.*, 2014) |
| Sodankyla, Finland | 67.4 | 26.6 | 188 | (Kivi *et al.*, 2014) |
| Tsukuba, Japan | 36.1 | 140.1 | 30 | (Morino *et al.*, 2014) |
| Wollongong, Australia | -34.4 | 150.9 | 30 | (Griffith *et al.*, 2014b) |

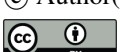



**Table 3. Summary of zonal mean biases of the forward and posterior simulations compared to MOPITT $X_{CO}$ and to ground measurements.** NH represent 30-90°N, Ntr represent 0-30°N, Str represent 0-30°S, and SH represent 30-90°S. Sub-columns in and tr represent results with the two OH: INCA or TransCom. The MOPITT $X_{CO}$ results here are sampled at corresponding ground stations instead of the entire latitudinal bands.

| | | MR | | | | HR1 | | | | HR2 | | | |
|---|---|---|---|---|---|---|---|---|---|---|---|---|---|
| | | Xco (ppb) | | Surface (ppb) | | Xco (ppb) | | Surface (ppb) | | Xco (ppb) | | Surface (ppb) | |
| | | F | P | F | P | F | P | F | P | F | P | F | P |
| HN | in | -23.6±9.9 | -0.7±4.4 | -36.0±20.7 | 27.6±39.2 | -20.4±9.0 | 2.0±4.7 | -34.9±30.5 | 27.6±45.7 | -20.9±9.2 | 1.4±5.1 | -33.2±30.2 | 32.5±47.2 |
| | tr | -14.9±7.2 | -0.8±4.1 | -24.9±21.4 | 18.2±23.8 | -11.8±6.9 | 1.9±4.3 | -24.0±30.7 | 16.6±33.8 | -12.1±7.1 | 1.9±4.5 | -22.1±30.5 | 21.2±34.5 |
| Ntr | in | -11.7±10.4 | -0.6±5.4 | -21.8±13.4 | -0.3±11.2 | -9.3±10.2 | 1.2±6.3 | -14.8±17.2 | 4.9±15.3 | -9.6±10.4 | 0.6±6.3 | -14.7±17.5 | 5.2±15.8 |
| | tr | -7.0±8.9 | -0.2±5.2 | -13.8±11.4 | 0.3±12.0 | -4.2±9.1 | 2.6±6.0 | -6.4±16.2 | 5.3±16.2 | -4.5±9.4 | 2.4±5.9 | -6.3±16.9 | 5.9±16.8 |
| Str | in | -7.1±7.7 | -0.5±4.5 | -7.6±10.1 | 3.3±15.5 | -8.3±9.2 | -1.3±6.0 | -5.3±21.9 | 1.8±23.1 | -7.5±8.8 | -0.7±5.7 | -4.8±22.8 | 2.0±23.1 |
| | tr | -8.3±9.1 | -0.4±4.6 | -8.9±9.7 | 5.5±17.8 | -8.0±10.8 | 0.1±5.8 | -4.8±21.9 | 4.0±24.1 | -7.4±10.5 | 0.9±5.4 | -4.5±23.0 | 4.3±24.1 |
| HS | in | -5.8±4.0 | 0.2±2.2 | 0.5±12.1 | 9.9±13.8 | -5.2±3.9 | 0.7±2.5 | 2.5±6.8 | 11.8±8.4 | -5.3±3.9 | 0.4±2.4 | 2.4±6.8 | 11.2±8.4 |
| | tr | -7.1±4.5 | 0.3±2.2 | -2.1±12.8 | 9.4±15.4 | -4.9±4.3 | 2.2±2.4 | 2.6±7.1 | 14.3±9.3 | -5.2±4.4 | 2.0±2.4 | 2.1±7.1 | 13.8±9.4 |





**Figure 1. Spatial distribution of the independent measurements.** The TCCON stations are shown by red triangles, the surface stations are shown by green circles, the MOZAIC/IAGOS and HIPPO profile projections on horizontal map are shown at the model resolution in orange and blue respectively.

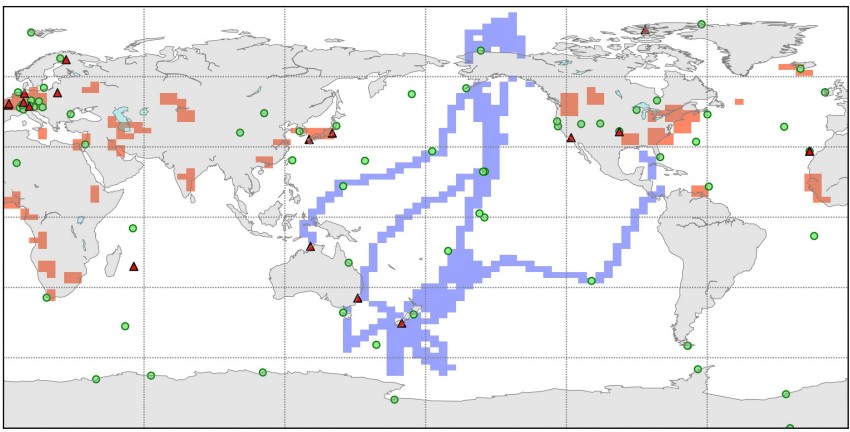




**Figure 2. Model-data biases along the latitudes. (a)** and **(b)** present model biases to MOPITT Xco using INCA-OH and TransCom-OH, respectively. Dashed and solid lines represent forward and posterior model results, respectively, while color codes correspond to model versions shown in the legend. **(c)** and **(d)** present model biases to TCCON Xco measurements using INCA-OH and TransCom-OH respectively. **(e)** and **(f)** show model biases compared to ground measurements from surface network using INCA-OH and TransCom-OH respectively. Note the ordinates are different for the three observation types.







**Figure 3. Covariation of model-data differences in the surface [CO] and in the MOPITT X$_{CO}$.** **(a)** and **(b)** show prior and posterior model results in the NH, **(c)** and **(d)** show prior and posterior model results in the SH.


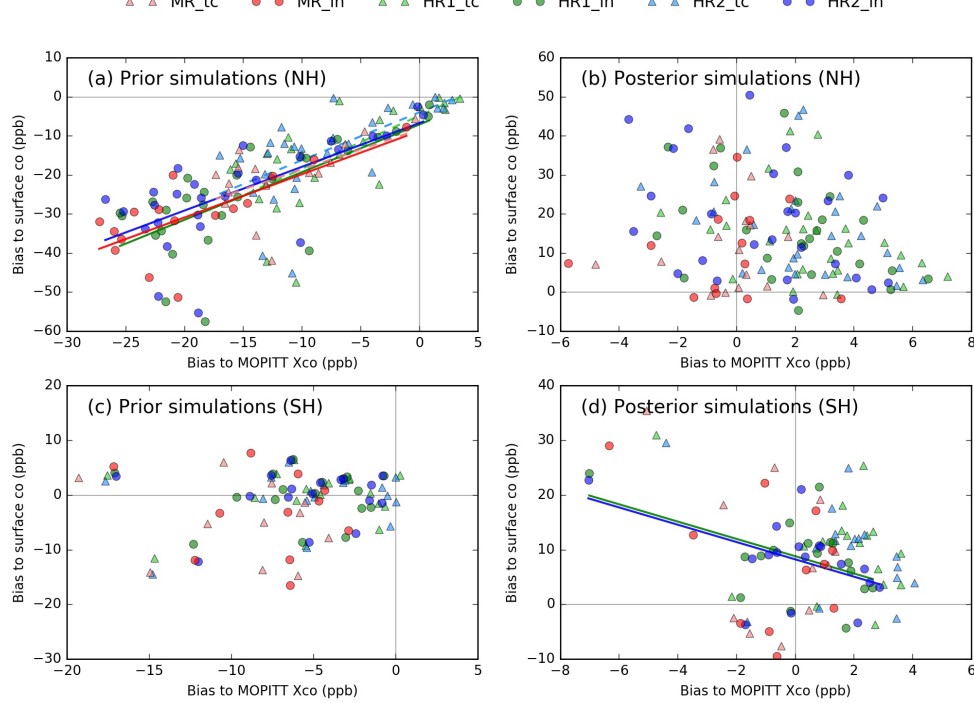




**Figure 4**. **Zonal averages of the vertical CO profiles.** The left column shows results against MOZAIC measurements, and the right column shows results against HIPPO measurements. The error bars indicate the standard variation of the measurements. Results associated with INCA-OH are presented.

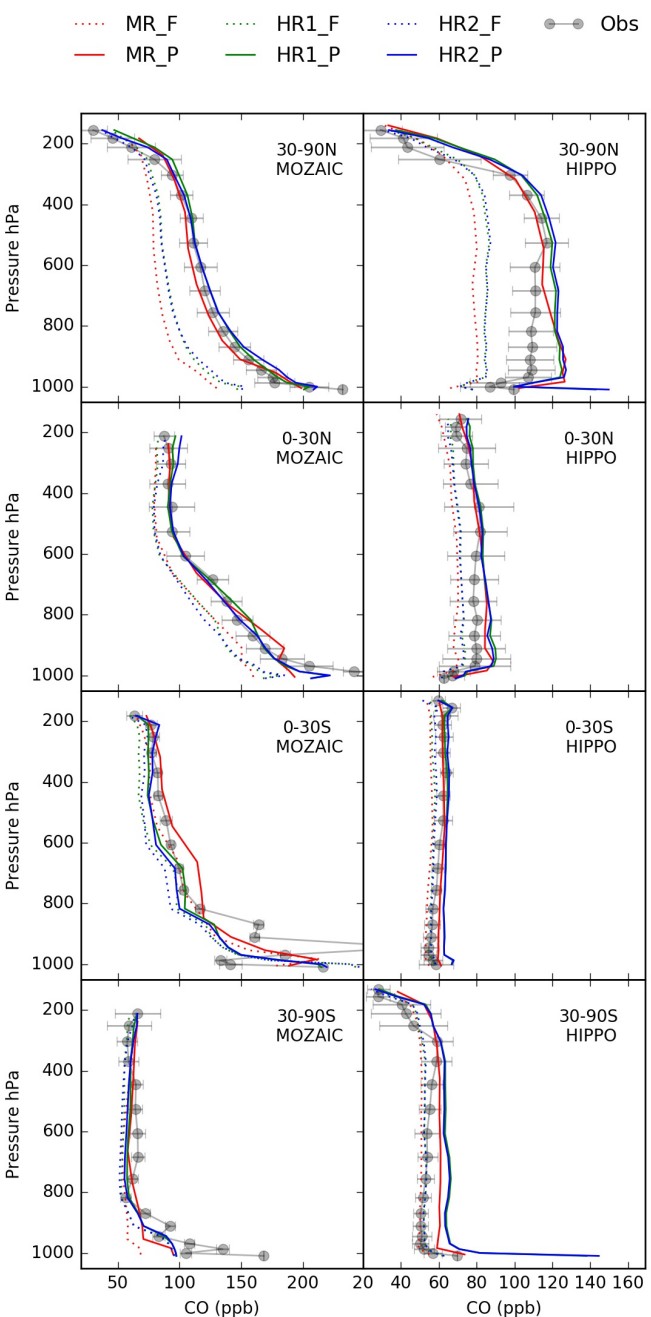





**Figure 5. Globally-average model-data biases of vertical profiles and vertical distribution of model-data correlation coefficients compared to MOZAIC and HIPPO measurements respectively.** The upper row shows results with INCA-OH, and the lower row shows results with TransCom-OH.

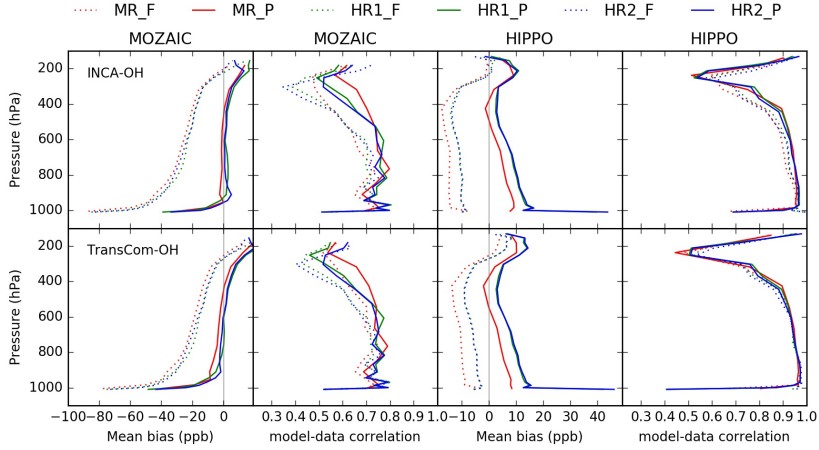





**Figure 6. Zonal averages of the vertical CO profiles at the MOPITT retrieval pressure**
**levels.** The upper panel successively shows the latitudinal average of the vertical profiles of **(a)**
the MOPITTv6 retrievals and prior forward model retrievals using **(b)** 19-level (MR_F_in) and **(c)**
39-level versions (HR1_F_in) with INCA-OH. The corresponding posterior model results are
shown below for visual comparison **(e)** MR_P_in and **(f)** HR1_P_in. The lower left subplot
shows **(d)** the zonal average of the vertical profiles of four bands, NH - North of 30°N, NTr - 0°-
30°N, STr - 0°-30°S, and SH - South of 30°S, where the solid lines represent the MOPITT profile
retrievals, the dotted-lines represent the forward simulation, and the dot-dash lines represent the
posterior model using INCA-OH. The results with TransCom-OH are of similar pattern, thus not
shown here.

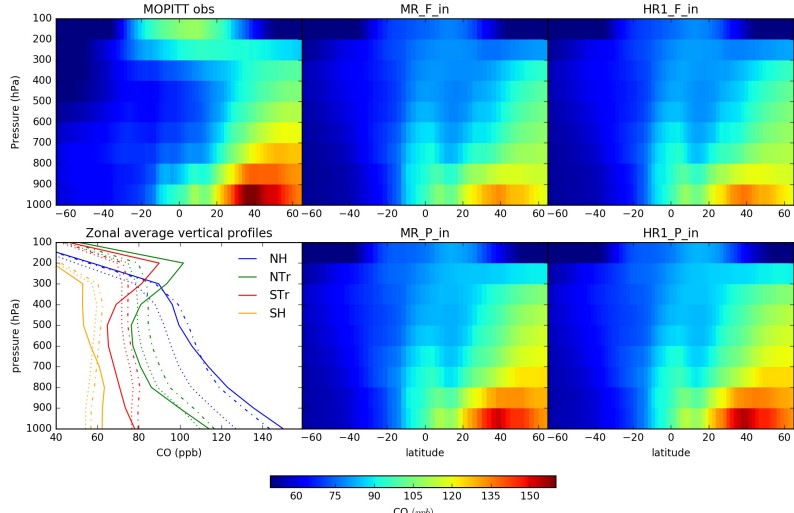




**Figure 7. Zonally-averaged model differences to MOPITT profile retrievals in (a) prior forward simulations and (b) posterior simulations.** Color codes for sub-regions are the same as Figure 6d. Solid lines represent the reference model with INCA-OH (MR_in), dotted lines represent the HR1 model with INCA-OH, dash-dotted lines show the HR1 model with
TransCom-OH. The differences between HR1 and HR2 given the same OH are not significant, and thus not shown.




**Figure 8. Pseudo Taylor diagrams of model agreement with MOPITT profiles for each region.** In each subplot, the radical distance indicates the relative bias (not standard deviation), with 1.0 being equal to the observation as shown by the dashed lines; the azimuthal angle is proportional to the model-data correlation coefficient, with higher correlations close to the horizon; the distance from to the reference data does not have an explicit statistical meaning, but qualitatively speaking the closer to the reference, the better it fits the observations. The hollow and filled markers represent forward and posterior simulations respectively, the color codes represent pressure level, the marker shapes represent model version.

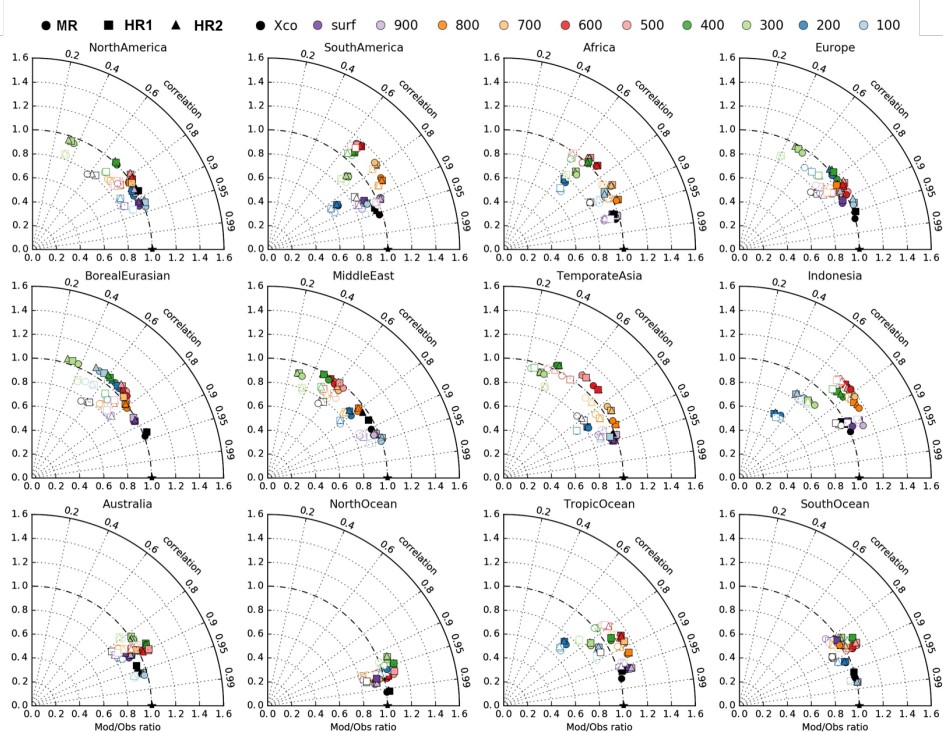




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
