# Peer review of "On biases in atmospheric CO inversions assimilating MOPITT satellite retrievals"

_Atmospheric Chemistry and Physics, 2017_

## Short Comment (SC1) · 6 Apr 2017

Comment by Helen Worden and Zhe Jiang, NCAR MOPITT team

We do not think the authors have demonstrated their conclusion that assimilating MO-PITT total column CO is superior to assimilating CO profiles since they did not show results from both assimilation types. Rather, they show comparisons to MOPITT retrieved profiles and other correlative data after assimilating the column quantities. These comparisons show biases already noted in MOPITT validation (Deeter et al., 2013;2014) and other assimilation results (e.g., Gaubert et al, 2016, Jiang et al., 2015;2016).

The paper states: "the number of retrieved layers exceeds the number of independent information available vertically". While this is true, the MOPITT CO profiles still have degrees of freedom for signal (DFS) > 1, (Deeter et al., 2015), where DFS=1 is the

most that can be obtained from a total column quantity. A method to assimilate only the independent layers (using singular value decomposition) along with the transformed full aposteriori error covariance is described in Mizzi et al., (2016).

In particular, the authors should demonstrate the following to substantiate their conclusion:

1) That assimilation results with MOPITT profile data, after bias corrections. are significantly worse than column assimilation as compared to independent CO observations.

2) That the vertical information (DFS $\sim$ 2.0 for profile vs. 1.0 for column) has no significant impact on the assimilation in correcting the vertical distribution of CO.

Added references: Deeter, M. N., D. P. Edwards, J. C. Gille, and H. M. Worden (2015), Information content of MOPITT CO profile retrievals: Temporal and geographical variability, J. Geophys. Res.-Atmos., 120(24), 12723–12738, doi:10.1002/2015JD024024.

Mizzi, A. P., Arellano, A. F., Edwards, D. P., Anderson, J. L., and Pfister, G. G.: Assimilating compact phase space retrievals of atmospheric composition with WRF-Chem/DART: a regional chemical transport/ensemble Kalman filter data assimilation system, Geosci. Model Dev., 9, 965-978, doi:10.5194/gmd-9-965-2016, 2016.
* * *

---

## Short Comment (SC2) · 17 Apr 2017

As listed below, the manuscript submitted by Yin et al. contains a number of errors with respect to the interpretation and treatment of the MOPITT data.

1. Section 2.1 includes several potentially serious errors. First, MOPITT total column values are not 'total column integrated dry air' values (line 135), since retrieved MO-PITT total column values quantify all of the CO molecules in a vertical column (per unit area), regardless of the moisture profile. Does this error affect the way $X_{co}$ is calculated? There are also several problems with respect to the authors' understanding of the MOPITT retrieval algorithm. Eq. 1 is not itself used in the MOPITT retrieval algorithm to calculate retrieved CO profiles or total column values, as the manuscript implies; really this equation just describes the expected relationship between the re-
trieved profile, a priori profile, and true atmospheric state in a 'maximum a posteriori' retrieval method. Eq. 1 contains a term A' which should actually be the total column averaging kernel (a vector) and not the averaging kernel matrix (line 144). Did the authors use the total column averaging kernel or the averaging kernel matrix in their calculations? Also, the term in parentheses in Eq. 1 generally represents the difference between the true atmospheric state and the a priori profile (rather than the difference in a model profile and the a priori). The paper also suggests that the x_mod term in Eq. 1 (which should be replaced by x_true) is somehow based on information gained from a 'radiance transfer model' (line 141). This is also incorrect. Calculation of the total column averaging kernel depends on assumed delta-pressure values for the MOPITT retrieval grid. The level-layer scheme which defines the delta-pressure values for V6 is described in the V5 validation paper (and V5 User's Guide). The level-layer scheme used in V5 and V6 products is the same, but is different than the scheme described in the V4 User's Guide. Did the authors use the proper level-layer scheme? Finally, it is not reported exactly how X_co is derived from C, the MOPITT CO total column.

2. The manuscript cites the MOPITT V6 validation paper, but does not include a review of the validation results for the V6 TIR/NIR product, and does not make use of those results when interpreting the posterior simulations. These earlier results represent the most direct method for quantifying the MOPITT retrieval bias and are clearly relevant to the work presented in this manuscript.

3. In Section 4, posterior simulations based on MOPITT X_co observations are compared with MOPITT retrieved profiles. However, for these comparisons, it is not clear whether or not the posterior simulations have been transformed with the 'observation operator' (i.e., x_sim = x_a + A(x_posterior - x_a) ), which is necessary to make a proper comparison.

---

## Referee Comment (RC1) · B. Gaubert (Referee) · 18 Apr 2017

General Comments: This paper aims to evaluate the representation of the CO fields obtained from the assimilation of MOPITTv6 total column (XCO) retrievals (one experiment) and several modelling sensitivity tests. The set of parameters from those sensitivity tests includes the use of posterior emission fields, OH fields, and transport through different grid spacing's, boundary layers and convection schemes.

Further research is needed in some studies using data assimilation to investigate, quantify and in fine understand biases from satellite retrievals, model simulation, and potentially from the assimilation methods (observation operator, error characterization, state vector choices). The comparison of analysis/posterior products can provide a different perspective for the nature of those biases, different from instrument validation and model evaluation by itself. In addition, the use of different and independent datasets is usually informative about the quality of the assimilated and/or posterior model fields. However, the paper's objectives, methods and conclusions need stronger clarifications. In other words, the conclusions can be misleading with regards to the scientific methods used. The authors should either review their conclusions or consider another evaluation approach before publication.

From the conclusions/abstracts, it is stated that the "purpose of top-down estimates of CO emissions, in which the model cannot directly correct vertical model biases, it is more robust to assimilate the column than a particular pressure level retrieval, a partial profile [. . .]".

As it is known, assimilating total columns presents some advantages. For instance, total columns can have lower instrument biases. For pragmatic reasons, it makes sense and I agree that if the observation does not give any information about the vertical distribution, it is more appropriate to not correct for the vertical distribution within the assimilation scheme. However, those conclusions are surprising with regards to the actual work that has been done in the paper. There are no tests that would rigorously test the hypothesis of choosing a method versus another. In addition, some comments are contradictory.

One important point to consider is that assimilating total columns relies on the vertical profile given by the model, thus, it is just the total column abundance that is shifted. Within that framework, there is no reason to expect improvement of the vertical profile, because it means that the modelled vertical profile is assumed to be perfect (relatively to the satellite observation). Or it can be assumed that it is far to be the case in global models (with coarse resolution in both vertical and horizontal). It is also acknowledged in the paper: "From the CTM perspective, the evaluation against aircraft measurements reveals significant model errors in representing the vertical CO gradient, in particular over the ocean."

Moreover, those errors persist after optimizing emissions, which means that chemistry and transport are both important. This is also acknowledged in the paper, but in the abstract, it is mentioned that: "Consistent negative prior biases to all types of observations in all sensitivity tests suggest an underestimation of current surface emissions in the Northern hemisphere. In contrast, prior simulations fit the surface air sample observations well in the Southern hemisphere but underestimate CO in the free troposphere and on average in the column."

For instance, Stein et al. (2014) demonstrated that the northern hemisphere spring cannot be attributed to direct CO emission alone. Myzaki et al. (2015) with a simple sensitivity test, a change in the CO + OH reaction rate were able to considerably reduce this bias. These studies suggest that this bias is likely to occur due to a combination of errors from chemistry, deposition, direct and indirect emission processes (vertical distribution, time profile), as well as transport. Far from the sources (in the free troposphere, over the ocean, in the southern hemisphere), it is even more likely that the problem will be due to transport and chemistry (secondary sources of CO and/or the OH sink). In particular, biases can be shared through the CH4/CO/OH system, as Strode et al. (2015) and Elshorbany et al. (2016) posit. In general, please consider discussing those previously published results with regards to the setup used in this study, which presents some advantages, but also some drawbacks to compare and contrast different possible approaches.

In order to improve the study please consider the following: 1. If this paper is aimed to evaluate the impact of assimilating either the profile or the total columns, this can be done using data assimilation experiments together with observation space diagnostics, see El Amraoui et al. (2014). The use of innovation statistics (and data assimilation) diagnostics allows to quantify the bias, while taking into account error variances from both model and observations. In particular, looking at those by Âň Assimilation of total columns and evaluation of profiles (in observation space) Âň Assimilation of profiles and evaluation of columns (in observation space)

By doing this the diagnostics for the assimilated and non-assimilated observations can be run for each case to verify the consistency between the columns and the profile in the observation space. In a case where assimilation parameters are correct (with regards to those same diagnostics) and underlying assumptions are not too violated, it would give a more robust estimate of the impact on the total columns while assimilating profiles and vice-versa.

2. Other studies that aim to improve the model error representation in chemical data assimilation (e.g., Gaubert et al. 2016; Emili et al. 2016 and reference therein) as well as potentials from strong constraint 4D-Var (e.g. Trémolet 2006) could be discussed and analyzed.

3. ÂăThe assimilation of compact phaseÂăspace retrievals (CPSRs) could be considered, which is an alternative approach to profile assimilation (e.g. Mizzi et al. 2016).

4. Thanks to the comparison to HIPPO measurements, the first identification of an upper tropospheric bias last from the MOPITT V4 (Deeter et al. 2010). The identification of the bias and update of the statistics against HIPPO has continued since then (Deeter et al., 2013; Deeter et al., 2017). You can review Martínez-Alonso et al. (2014) for another evaluation of MOPITT profiles and satellite data. Jiang et al. (2013) suggest not to assimilate the profiles in the upper troposphere while Jiang et al. (2017) propose a latitudinal bias correction. There is evidence that errors can arise from the multispectral retrieval for the nighttime oceanic scene (Worden et al., 2010).

5. Finally, please consider to include an extensive discussion on the sources of model error and limitation of the setup of the simulations. You can provide more quantitative results. To do so, it is highly recommendable to use conservative measures to compare simulations such as atmospheric burden for CO abundance and RMSE (or absolute errors) for comparison of simulation with observations. On the spatial resolution, one can also discuss the fact that emissions estimates would be different at higher resolution. The change of resolution changes the distribution of emissions and

thus chemical regimes. The errors evolution with regards to spatial grid spacing can be not linear in chemical transport models. Here, the emissions have been optimized at lower resolution, this leads to serious limitation to conclude about the effect of spatial resolution.

6. Perhaps, a further discussion on the limitation of the methods is needed, knowing that emission is not the only error source. For instance, there is dry deposition (Stein et al. 2014), time evolution coupling and feedbacks of chemistry (Strode et al., 2015; Gaubert et al. 2016; Elshorbany et al., 2016) and vertical transport, the aggregation of VOC's oxidation in one term (e.g. Jiang et al., 2015 and reference therein), etc.

References

Barré, J., et al. (2015), Assessing the impacts of assimilating IASI and MOPITT CO retrievals using CESM-CAM-chem and DART, J. Geophys. Res. Atmos., 120, 10,501–10,529, doi:10.1002/2015JD023467.

Deeter, M. N., et al. (2010), The MOPITT version 4 CO product: Algorithm enhancements, validation, and long-term stability, J. Geophys. Res., 115, D07306, doi:10.1029/2009JD013005.

Deeter, M. N., S. Martínez-Alonso, D. P. Edwards, L. K. Emmons, J. C. Gille, H. M. Worden, J. V. Pittman, B. C. Daube, and S. C. Wofsy (2013), Validation of MOPITT Version 5 thermal-infrared, near-infrared, and multispectral carbon monoxide profile retrievals for 2000–2011, J. Geophys. Res. Atmos., 118, 6710–6725, doi:10.1002/jgrd.50272.

Deeter, M. N., Edwards, D. P., Francis, G. L., Gille, J. C., Martinez-Alonso, S., Worden, H. M., and C. Sweeney (2017), A Climate-scale Satellite Record for Carbon Monoxide: The MOPITT Version 7 Product, Atmos. Meas. Tech. Discuss., doi:10.5194/amt-2017-71, in review.

El Amraoui, L., Attié, J.-L., Ricaud, P., Lahoz, W. A., Piacentini, A., Peuch, V.-H., Warner, J. X., Abida, R., Barré, J., and R. Zbinden (2014) Tropospheric CO vertical

profiles deduced from total columns using data assimilation: methodology and validation, Atmos. Meas. Tech., 7, 3035-3057, doi:10.5194/amt-7-3035-2014.

Elshorbany, Y. F., Duncan, B. N., Strode, S. A., Wang, J. S., and Kouatchou, J.: The description and validation of the computationally Efficient CH4–CO–OH (EC-COHv1.01) chemistry module for 3-D model applications, Geosci. Model Dev., 9, 799-822, doi:10.5194/gmd-9-799-2016, 2016.

Fisher, J. A., Wilson, S. R., Zeng, G., Williams, J. E., Emmons, L. K., Langenfelds, R. L., Krummel, P. B., and Steele, L. P.: Seasonal changes in the tropospheric carbon monoxide profile over the remote Southern Hemisphere evaluated using multi-model simulations and aircraft observations, Atmos. Chem. Phys., 15, 3217-3239, doi:10.5194/acp-15-3217-2015, 2015.

Jiang, Z., D. B. A. Jones, H. M. Worden, M. N. Deeter, D. K. Henze, J. Worden, K. W. Bowman, C. A. M. Brenninkmeijer, and T. J. Schuck (2013), Impact of model errors in convective transport on CO source estimates inferred from MOPITT CO retrievals, J. Geophys. Res. Atmos., 118, 2073–2083, doi:10.1002/jgrd.50216.

Jiang, Z., Worden, J. R., Worden, H., Deeter, M., Jones, D. B. A., Arellano, A. F., and Henze, D. K. (2017), A 15-year record of CO emissions constrained by MOPITT CO observations, Atmos. Chem. Phys., 17, 4565-4583, doi:10.5194/acp-17-4565-2017.

Mizzi, A. P., ArellanoÂăJr., A. F., Edwards, D. P., Anderson, J. L., and Pfister, G. G.: Assimilating compact phase space retrievals of atmospheric composition with WRF-Chem/DART: a regional chemical transport/ensemble Kalman filter data assimilation system, Geosci. Model Dev., 9, 965-978, doi:10.5194/gmd-9-965-2016, 2016. Strode, S. A., B. N. Duncan, E. A. Yegorova, J. Kouatchou, J. R. Ziemke, and A. R. Douglass (2015), Implications of carbon monoxide bias for methane lifetime and atmospheric composition in chemistry climate models, Atmos. Chem. Phys., 15, 11,789–11,805, doi:10.5194/acp-15-11789-2015. Trémolet, Y.: Accounting for an imperfect model in 4D-Var, Q. J. Roy. Meteor. Soc., 132, 2483–2504, doi:10.1256/qj.05.224, 2006. Worden, H. M., Deeter, M. N., Edwards, D. P., Gille, J. C., Drummond, J. R., and Nédélec, P. (2010), Observations of near-surface carbon monoxide from space using MOPITT multispectral retrievals, J. Geophys. Res., 115, D18 314, doi:10.1029/2010JD014242, http://doi.wiley.com/10. 1029/2010JD014242.

Specific Comments:

There are some sentences starting with "It seems" followed by a strong fact statement: Âň "The choice of the prior OH seems to produce the largest differences in the simulated CO concentrations at a global scale." Âň "representing the vertical profile correctly seems to be a grand challenge." This is confusing, please choose between hypothesis or fact.

The Section "2.1 MOPITT satellite retrievals of CO total column and vertical profiles" is not clear and contains errors. It needs major revision. Can you merge with section 2.4.4? I don't think the authors should presents the retrieval algorithms. Please provide instead the way they applied the averaging kernels for the columns (equation 1) and the profile (equation 2).

From Yin et al. 2015, it is stated that the MOPITT observations are average on the coarser grid. For a fair comparison, the evaluation should be done at 'higher' resolution, note that high resolution for a simulation at 1.895 by 2.5 degrees is misleading. A comparison of the impact of the vertical grid spacing should be done first, what is the observation error differences between low ($2.5°\times3.75°$) and medium (1.895 by 2.5 degrees)? What is the differences in biases?

Minor Comments

Abstract

Page 1, Line 16: "Carbon monoxide (CO) inverse modelling studies have so far reported significant discrepancies between model concentrations optimized with . . . (MOPITT) satellite retrievals and surface in-situ measurements."

In my opinion, this first sentence is misleading and is not fair to MOPITT itself. • It is common in atmospheric composition that there are discrepancies satellite and surface observations, or between satellite themselves (Kopacz et al. 2010). • It usually leads to large improvements in particular for the well-known northern hemisphere spring bias. Which means that the errors are also due to model and coarse resolution. • There have been issues in the Southern Hemisphere, but the reasons are not clear. The model errors appear to not be driven by CO emission alone (e.g. Fisher et al., 2015).

1 Introduction

P2, L76: "with most CTMs showing negative biases to surface and satellite observations in the Northern hemisphere when prescribed with current emission inventories (Naik et al., 2013; Patra et al., 2011; Shindell et al., 2006)." The Patra et al.'s study is about TransCom-CH4 and the paragraph is CTM's biases in Northern Hemisphere spring of CO modelling, please change this reference to Stein et al. (2014).

P2, L85: "MOPITT-based atmospheric inversions were also shown to be biased high when compared to independent in-situ surface observations in the boundary layer (Gaubert et al., 2016; Yin et al., 2015)." Gaubert et al. (2016) showed that assimilating MOPITT improve the CO values at the surface (not the opposite). The cross-validation with FTS in the southern hemisphere suggests that the model has actually a good prior for wrong reasons, the assimilation improve correlation and suggest an underestimation of biomass burning emissions. Please be more precise, at least indicate that it is in the southern extra-tropical region.

P3, L107: "In section 3, we first evaluate the model with MOPITT XCO that is used for assimilation, then [. . .]". Please clarify, only one simulation is done with assimilation of MOPITT XCO, what is the model?

P4 L152: change "coverting" to converting

P4, L186: "HR2 corresponds to the version called standard physics (SP) in (Locatelli

et al., 2015), where more detail regarding these configurations are described." Please rephrase, e.g. "The latter corresponds to a version called standard physics (SP), presented in Locatelli et al. (2015), where more details about these configurations are described."

P5, L189: Please rephrase, "as suggested by Radon [...]", to "as suggested by the comparison with Radon [...]".

P5, L191: Please rephrase "For all versions of LMDz-SACS here", For all the different LMDz-SACS configurations presented in this study: [...] boundary conditions and horizontal winds [...]"

P5: L214: "One scale factor is applied to the HR results over the globe to conserve the global mass budget to be consistent with the reference version MR." Can you explain and be more explicit? If the 3D fields of Formaldehyde have been simulated using different spatial resolution, why would not you keep a different field? Which budget (CO or HCHO)?

P6, L269: "It is noted that the MOPITT NIR/TIR retrievals, combining information from both TIR and NIR, have generally higher sensitivity to the lower troposphere compared to TCCON." Please rephrase, add a reference?

P7 L305: "Individual aircraft profiles are assigned to certain model grid points given their geographic location and pressure levels, and all measurements are then averaged per model grid point at a 30-minute resolution to compare with corresponding model value." What would be the impact of spatial resolution? This comparison is in favor of the coarser resolution, why don't you interpolate to the observations?

P7 L327: I do not understand this statement: "the total column of the model state integral is always conserved to compensate uncertainty from vertical resolution change on the CTM side." Again, the evaluation at lower resolution is in favor of the coarse run.

P8 L341: Typo, "of the model version or of the OH field"

P8 L345: change "for both OH" to "for both OH fields".

P8 L350: "It is noted that although we optimize OH together with surface emissions, the system only scales slightly the big-region OH state vectors (in total six big regions over the globe) and thus the inverted surface emissions are sensitive to the prior OH fields." With OH driving the CO sinks (90 %), how would it be possible to not be sensitive to the prior OH field?

P8 L350: "with the range showing 1-sigma standard deviation of the mean biases of all model grids, the same applies hereafter if not specified otherwise". Please remind the time period, a large variability is expected for the northern hemisphere spring. P8 L365: "The CO surface sources are conserved in global mass between different resolutions when emitted to the atmosphere, but with the change in resolution, the CO sink via the reaction with OH (associated with different resolutions) may differ." What is the purpose of changing the resolution if you forced to value to be equal to the coarse resolution? Those differences are what can be interested in this study, because you are comparing simulation with different spatial grid spacing.

P8 L370: "The difference between the HR and MR XCO results ($\sim$1.5 ppb) is of a much smaller magnitude than the differences between the prior and the posterior MR simulations ($\sim$10.4 ppb); it is also of a smaller magnitude than that induced by the two OH fields in the prior forward simulations when the CO sources are identical ($\sim$2.8 ppb for global average with some cancelling effect between the NH and SH). The differences in modelled XCO between HR1 and HR2 are not significant at a global scale." To remove cancelling effects, could you use RMSE for comparison with measurements, and/or tropospheric abundance (tropospheric mass) for the comparison of different simulations.

P9 L388: "They also slightly overestimate XCO in SH, resulting in a smaller positive bias 2.5$\pm$3.1 ppb using INCA-OH (3.0$\pm$2.2 ppb using TransCom-OH)." Please rephrase.

P9 L422: "when other setups being the same; here, the 1s standard deviation showing

the spread across prior/posterior simulations and between the two HR models." Please do not use the average difference, is it with adjusting emissions and chemistry. You could also do a simulation with full chemistry and updated emissions.

P10 L430 and Figure 3: What are the different points, is it the latitude band or the time period (Months)? A large seasonal cycle is expected (see https://www.esrl.noaa.gov/gmd/ccgg/globalview/co/co_intro.html).

P10 L449 and P10 L469: title 3.4.1 MOZAIC measurements over large airports What do you mean by large airports? Did you actually select airports that are larger, in size, in number of flights?

P10 L455: "are larger in the NH than in the SH, consistent with the different in prior model" Please correct to 'differences' or 'different priors'.

P11 L511: "Such bias in representing the oceanic vertical profiles suggests error in the vertical distribution of CO source/sink over ocean or in the vertical mixing." Please rephrase, there are errors (plural) from both chemistry, horizontal and vertical transport, ocean is repeated twice.

P13 L600: "The choice of the prior OH seems to produce the largest differences in the simulated CO concentrations at a global scale." It is confusing, please remove 'seems', or you can say "Our study shows that the choice of. . ." Again, do not forgot that having a prescribed OH is a strong approximation (limitation), see for instance (Elshorbany et al.; 2016)

P13 L615: "The North-to-South gradient in TransCom-OH is also closer to a recent observation-derived near equal N/S OH distribution (Patra et al., 2014)".

I think those two studies are related, the TransCom-OH field is from ACTM_0.99, which is designed to have a near equal N/S OH distribution. You can mention this in the introduction, it is supposed to be the best fit to Methyl Chloroform.

Acknowledgement Most of the acknowledgement are not respecting the recommen-

dation, please consider contacting the instrument PI's. For instance, the MOZAIC acknowledgement (if not updated since) should be as follow:

The authors acknowledge the strong support of the European Commission, Airbus, and the airlines (Lufthansa, Air France, Austrian, Air Namibia, Cathay Pacific, Iberia and China Airlines so far) which have carried the MOZAIC or IAGOS equipment and undertaken maintenance since 1994. In its last 10 years of operation, MOZAIC has been funded by INSU-CNRS (France), Météo-France, Université Paul Sabatier (Toulouse, France) and Research Center JuÌĹlich (FZJ, JuÌĹlich, Germany). IAGOS has been additionally funded by the EU projects IAGOS-DS and IAGOS-ERI. The MOZAIC–IAGOS database is supported by AERIS (CNES and INSU-CNRS).

Table 1: "List of simulations in this study", -> 'list of simulations done in this study' or 'list of simulations'.

Table 2: "reference" to "references"

Table 3: How are calculated the error bars (more or less)?

---

## Short Comment (SC3) · 22 Apr 2017

The author's reply to our comment did not address the serious deficiency that their study does not provide evidence to support their conclusions about assimilating profile vs. column CO MOPITT retrievals, since they did not assimilate profile data, even without bias correction. It would be necessary (but still not sufficient) to show that the assimilation of profile data, without bias corrections, performs significantly worse than column assimilation as compared to independent CO observations. I disagree with their assessment that MOPITT retrieval biases cannot be corrected due to their spatial and temporal variability. Although this correction could be tedious, and it will likely be less accurate where there is a lack of validation data, bias information is provided in Deeter et al., (2014) and Buchholz et al., (2017) for MOPITT V6 and Deeter et al., (2017) for MOPITT V7. After demonstrating the performance of the profile assimilation, if their conclusion about assimilating column CO vs. profiles is particular to the vertical biases in their model, they should make this more explicit in the abstract and conclusions rather than the blanket recommendation for assimilation only of column CO.

References: Deeter, M. N., Edwards, D. P., Francis, G. L., Gille, J. C., Martinez-Alonso, S., Worden, H. M., and Sweeney, C.: A Climate-scale Satellite Record for Carbon Monoxide: The MOPITT Version 7 Product, Atmos. Meas. Tech. Discuss., doi:10.5194/amt-2017-71, in review, 2017.

Buchholz, R. R., Deeter, M. N., Worden, H. M., Gille, J., Edwards, D. P., Hannigan, J. W., Jones, N. B., Paton-Walsh, C., Griffith, D. W. T., Smale, D., Robinson, J., Strong, K., Conway, S., Sussmann, R., Hase, F., Blumenstock, T., Mahieu, E., and Langerock, B.: Validation of MOPITT carbon monoxide using ground-based Fourier transform infrared spectrometer data from NDACC, Atmos. Meas. Tech. Discuss., doi:10.5194/amt-2016-241, accepted, 2017.

---

## Author Comment (AC1) · 22 Apr 2017

We thank Dr. Helen Worden and Dr. Zhe Jiang for their comments. They are discussed hereafter.

> *We do not think the authors have demonstrated their conclusion that assimilating MOPITT total column CO is superior to assimilating CO profiles since they did not show results from both assimilation types. Rather, they show comparisons to MOPITT retrieved profiles and other correlative data after assimilating the column quantities.*

Our demonstration is about the importance of retrieval and model biases for the CO profiles when assimilating CO retrievals to infer CO surface fluxes. Indeed, these model

and retrieval biases vary with the altitude and are poorly (retrievals) or not (model) constrained for this particular application (atmospheric inversion).

> *These comparisons show biases already noted in MOPITT validation (Deeter et al., 2013;2014) and other assimilation results (e.g., Gaubert et al, 2016, Jiang et al., 2015;2016).*

We already cited these papers that have helped setting the stage.

> *The paper states: "the number of retrieved layers exceeds the number of independent information available vertically". While this is true, the MOPITT CO profiles still have degrees of freedom for signal (DFS) > 1, (Deeter et al., 2015), where DFS=1 is the most that can be obtained from a total column quantity. A method to assimilate only the independent layers (using singular value decomposition) along with the transformed full aposteriori error covariance is described in Mizzi et al., (2016).*

We are well aware of this paper and of the underlying Migliorini papers that we cited in another study. However, this strategy (developed in the context of a 6-hour data assimilation system for a regional model) does not help here for a typically 1-year global inversion system that can change the vertical structure of the modelled profiles only marginally.

> *In particular, the authors should demonstrate the following to substantiate their conclusion: 1) That assimilation results with MOPITT profile data, after bias corrections. are significantly worse than column assimilation as compared to independent CO observations.*

Our point gets hidden behind the words "after bias corrections". Retrieval and model biases vary in time and 3D space (as shown here and in the studies mentioned above) and are not well constrained, or not constrained at all.

> *2) That the vertical information (DFS ~ 2.0 for profile vs. 1.0 for column) has no significant impact on the assimilation in correcting the vertical distribution of CO.*

We do appreciate the value of profile retrievals. However, in atmospheric inversions, the state vector is mainly made of surface CO fluxes, that are not enough to significantly change the vertical distribution of CO to the needed extent (as shown in Figure 4, the vertical gradients are similar between the prior and the posterior simulations even though the surface emissions are significantly changed). The situation is different for Numerical Weather Prediction (NWP)-type applications where the target quantity is the profile itself.

---

## Author Comment (AC2) · 22 Apr 2017

We thank Dr. Merritt Deeter for his remarks that will help clarify our text in the revised version. They will all be accounted for and are individually discussed hereafter.

> *As listed below, the manuscript submitted by Yin et al. contains a number of errors with respect to the interpretation and treatment of the MOPITT data.*

In practice, none of the comments contradict our treatment and interpretation of the MOPITT data, but the text will be changed to reflect this better.

> *Section 2.1 includes several potentially serious errors. First, MOPITT total column values are not 'total column integrated dry air' values (line*

[Figure]

*135), since retrieved MOPITT total column values quantify all of the CO molecules in a vertical column (per unit area), regardless of the moisture profile. Does this error affect the way $X_{co}$ is calculated?*

Our expression was awkward indeed and will be corrected. We have treated $X_{co}$ as the number of CO molecules in a vertical column per unit area (in unit $molec/cm^2$), regardless of the moisture profile.

*There are also several problems with respect to the authors' understanding of the MOPITT retrieval algorithm.*

The wording did not express our understanding well and will be improved.

*Eq. 1 is not itself used in the MOPITT retrieval algorithm to calculate retrieved CO profiles or total column values, as the manuscript implies; really this equation just describes the expected relationship between the retrieved profile, a priori profile, and true atmospheric state in a 'maximum a posteriori' retrieval method. Eq. 1 contains a term A' which should actually be the total column averaging kernel (a vector) and not the averaging kernel matrix (line 144).*

We agree and will change the paragraph accordingly.

*Did the authors use the total column averaging kernel or the averaging kernel matrix in their calculations?*

We used the total column averaging kernel for the total column and the averaging kernel matrix for the profile retrievals.

*Also, the term in parentheses in Eq. 1 generally represents the difference
between the true atmospheric state and the a priori profile (rather than the
difference in a model profile and the a priori). The paper also suggests that
the x_mod term in Eq.1 (which should be replaced by x_true) is somehow
based on information gained from a 'radiance transfer model' (line 141).
This is also incorrect.*

We will change the text accordingly.

*Calculation of the total column averaging kernel depends on assumed
delta-pressure values for the MOPITT retrieval grid. The level-layer scheme
which defines the delta-pressure values for V6 is described in the V5 val-
idation paper (and V5 User's Guide). The level-layer scheme used in V5
and V6 products is the same, but is different than the scheme described in
the V4 User's Guide. Did the authors use the proper level-layer scheme?*

Yes. We updated the delta-pressure values following User's Guide V5. But the conver-
sion between A and A' was documented in User's Guide V4, that's why User's Guide
V4 is cited. We did not think this point could cause any confusion, as by default, the
pressure grid should match the corresponding product. We will add this information
regarding delta-pressure values in the revised version.

*Finally, it is not reported exactly how $X_{co}$ is derived from C, the MOPITT CO
total column.*

As it is detailed in the User's Guide V4, section 7.4 (page 20-21), we considered it not
necessary to duplicate the information. We will add more detailed references in the
revised version, if it helps clarifying things.

*The manuscript cites the MOPITT V6 validation paper, but does not include a review of the validation results for the V6 TIR/NIR product, and does not make use of those results when interpreting the posterior simulations. These earlier results represent the most direct method for quantifying the MOPITT retrieval bias and are clearly relevant to the work presented in this manuscript.*

We cited the MOPITT papers that we consider most relevant and latest - in total four papers by Deeter et al.

*In Section 4, posterior simulations based on MOPITT $X_{co}$ observations are compared with MOPITT retrieved profiles. However, for these comparisons, it is not clear whether or not the posterior simulations have been transformed with the 'observation operator' (i.e., x_sim = x_a + A(x_posterior - x_a) ), which is necessary to make a proper comparison.*

It is calculated exactly as you suggest here, which is described in Section 2.4.4. line 315 – 318. We put it into a different section than the one describing the data assimilation, in order to make it clear that the vertical profiles are not directly assimilated. In fact, it was the reason we modified the expression compared to the original MOPITT documents (e.g. x_true to x_mod) to reflect how we calculate corresponding CTM retrievals in a consistent manner.

---

## Short Comment (SC4) · 27 Apr 2017

In general, the authors appear to understand the points I raised in my first comment, and have stated their intention to clarify these points in the revised version. I will therefore need to read the revised version before commenting further.

However, the authors do not seem to have understood the point I raised about the V6 TIR/NIR validation results. Their response to this comment was "We cited the MOPITT papers that we consider most relevant and latest - in total four papers by Deeter et al." The issue is not whether the V6 validation paper was cited or not. The point here is that any conclusions made in the submitted manuscript regarding retrieval bias (for example, in Sections 3.2, 3.2, and 5.3) should be related to what is already known about such biases. There is valuable information in the V6 validation paper, especially

regarding biases in the V6 TIR/NIR product over the Northern Hemisphere, that is completely ignored in the submitted manuscript.

---

## Referee Comment (RC2) · Anonymous Referee #2 · 15 May 2017

General comments:

The submitted paper aims to evaluate the quality of CO emissions inversions and inference of other parameters influencing the CO budget using different model configurations, i.e. horizontal and vertical resolution, OH forcing and physics scheme. While the authors try to provide a detailed evaluation, and provide indications of possible biases on MOPITT retrievals and additionally of model and emissions, the methodology and scientific argumentation employed is not sound to me. It is quite unclear what are the exact goals of the paper and it seems that the focus is too broad. The evaluation is extensive which is appreciable but the discussions try to cover too many topics without robustness and convincing arguments. This study needs to mature and needs to be supported by additional experiments. I fear I cannot recommend this paper for publication for the following main reasons:

[Figure]

1. The discussion and especially sections 5.3 and 5.4 are scientifically flawed. The authors drive conclusions without experimenting themselves. I strongly recommend the authors to reconsider their data assimilation experiments and setups before driving such conclusions or consider removing those two sections. I fear that without those two sections the paper will significantly loose substance. Moreover, the sensitivity tests on model parameters are not convincing, a significant increase on model horizontal resolution and using a more detailed chemical scheme would have been more useful to point out intrinsic model deficiencies and uncertainties.

2. The quality of the scientific argumentation can be questioned. A lot of references are cited inappropriately. Number of citations do not support statements made in the present paper (see specific comments below). Demonstrations are often approximate and hand-waving. The conditional form is often used when it comes to conclusions (the forms "would" and "could" are widely used). The authors suggest and anticipate from incomplete set of experiments with few references to drive scientific conclusions.

3. Last but not least, I am concerned about the methodology itself; statistical methodology and the significance of the diagnostics. The reliability of the data assimilation algorithm is not discussed as well.

To support the three mentioned points please consider the following specific comments.

Specific comments:

Line 66: There are also other references that are using MOPITT and data assimilation to study the temporal distribution and variability of CO, e.g. Inness et al., 2015, Myazaki et al., 2015, Barré et al., 2015.

Line 75: Please, change plagued by another word. Models are not plagued, they just misrepresent the truth by man-made simplifications.

Line 83-84: This is not what Hooghiemstra et al., 2012 are proving. Form the conclusion of the paper it is: "However, in the remote SH (30 – 60Âă S), the comparison

with MOPITT deteriorates from a 4% negative bias in the a priori to a 10% negative bias in the a posteriori solution, due to an emission decrease suggested by SH surface observations."

Line 86: Gaubert et al., 2016 is not inverting surface emissions as Yin et al., 2015.

Lines 84-87: This statement is not well supported by either reference provided. For example, Barré et al., 2015 that is assimilating two types of sounders find opposite conclusions. MOPITT assimilation still underestimates CO at the surface over CONUS. This is probably only true in the southern hemisphere.

Lines 87-90: This statement is not clear at all. Please clarify.

Line 88: differences between what and what?

Line 90: While the statement is unclear to me, I do not think this is the Jiang et al., 2015 conclusions.

Line 135: The authors should know what Bayesian means. There is nothing Bayesian in this equation.

Line 140: Clarify the statement, it sounds as you model a profile from measurements.

Line 176: 2.5 by 3.75 degrees is now considered as low resolution, change accordingly.

Line 179: is it another model? I believe you still use LMDz but with a different configuration. Change accordingly.

Line 181: Does changing just the latitudinal resolution from 2.5 to 1.89 degrees relevant? It is then mainly a significant increase on the vertical resolution. Why not keeping the same horizontal resolution? Again 1.89 by 3.75 by 39 levels is not considered nowadays as high resolution.

Lines 199-200: 2009 to 2011? From what month to what month? It could be almost three years to almost one year though.

[Figure]

Lines 316-317: The authors should detail exactly how they apply the observation operator to retrieve Xmod. Have they smoothed the model profile by the averaging kernel, have they considered interpolating partial columns from the model and then convert to log(vmr) to match the MOPITT data? The authors should refer to Barré et al., 2015 section 2.2.4 for the correct approach. I am then uncertain if the method used by the author is the correct one, hence I am doubtful about the validity of the results and discussion about the MOPITT profiles validation in rest of the paper.

Lines 323-324: Does this mean that you are taking the nearest grid point. If yes, is that appropriate? Since you are doing DA science you should be able to interpolate at the right location.

Lines 325-328: This is unclear to me, what operation the authors are doing here. Are you shifting or scaling the profile in order to keep the same total column value? What is the "uncertainty from vertical resolution change on the CTM"? Please rewrite, develop, explain better.

Lines 330-333: It is unclear to me what the authors are doing exactly. Are they averaging monthly model values and then they are comparing with monthly averaged observations? If yes, the entire results of this paper would be flawed.Or are they interpolating model to observation at the right time. Moreover, it seems that the correlations in the rest of the paper are made on monthly averaged biases, reducing the sample for correlation to something small and probably not statistically significant. Looking at the correlations plots I see around 12 to 14 point as a sample size. Would it be statistically more sound to calculate those correlations using the entire sample of observations (not reduced by average biases)? I am then doubtful about the robustness of this score during the further analysis of this paper.

Line 358: Please recall what are those big-regions. Cite Yin et al., 2015.

Line 370-376: This entire paragraph is confusing to me, please rewrite.

Line 427: Higher sensitivity of what to what?

Line 432-433: Please see my comment above about the significance and robustness of those correlations. The statistical methodology as it is presented now is not sound to me.

Line 443-445: This is again confusing with a "hand-waving" argument displaying only the HR correlations, using the word "likely" and not further investigating the possible error on the vertical error CO profile on the posterior. Additionally, I would not trust a correlation of 0.49 with a sample size of 14 using monthly averaged biases.

Line 469: The word contamination is not appropriately used. For example, there is contamination in data when an instrument is not working correctly and generate a systematic error. Please replace this word.

Lines 474-479, lines 506-513, figures 4 and 5: MOZAIC profiles are most likely close to the sources whereas HIPPO measures remote scenes. The bias observed in the posterior for HIPPO profiles are due to an overly long CO lifetime in the simplified chemistry model. The presented data assimilation system infers the surface CO sources but do not directly corrects for CO lifetime error due to an (over) simplified chemical scheme.

Lines 531-532: Please rephrase. "The MOPITT profiles are well reproduced by the model...". The model does not reproduce MOPITT profiles.

Lines 541-544: In Deeter et al., 2014, the MOPITT V6 validation with HIPPO do not see such errors in the upper troposphere. Also, the author should also take into account the spatial sampling of MOZAIC, HIPPO versus MOPITT. The longitudinal distribution of CO, in the tropics can be highly variable.

Line 565: Which tropical ocean? Rephrase.

Line 574: "over the ocean"

Line 607: What are those big regions? Recall or cite Yin et al., 2015.

[Figure]

Lines 638-643: What is the purpose of this paragraph? It is not clear what the authors are trying to demonstrate. Please clarify, develop, rephrase.

Lines 649-652: The syntax of this sentence is not correct.

Lines 683-695 and section 5.3 in general: The conclusion of "positive biases in the MOPITT retrievals" is flawed here. The authors utilize only one inversion technique from only using total column product. They infer only the surface emissions that is not a direct measured quantity from MOPITT CO retrievals. Depending of a model quality (i.e. resolution, chemistry, horizontal and vertical transport, and so on. . .) inverting the emissions only can lead to good result for the wrong reasons and conversely often having the "correct" emissions and having significant errors in the atmosphere. Data assimilation rely on observation but ALSO on models, you could have the best observation quality, if the model is inaccurate the analysis and the subsequent forecasts would be degraded. Before jumping quick in such important conclusions several things should be tested carefully such as:

Assimilating the CO fields directly with total CO columns and CO profiles Rerunning the current experiments with a more complete and detailed chemistry

Line 692-695: Deeter et al., 2014 made the direct comparisons between MOPITT V6T (which is the same as MOPITT V6J over the ocean) and HIPPO measurements: providing a quantification of the MOPITT biases: 1.5% 7.7% at the 200hPa level. How can the authors can explain such discrepancies with those results and figure 4 and 7. The authors compare figure 2 and 4 with figure 7. Again, the representativity of the statistics made here should be considered. HIPPO and MOZAIC cover specific regions whereas MOPITT provide a global picture. Is it reasonable to compare those figures in order to drive conclusions about biases without quantification?

Lines 702-707: This indicate an issue in your CO lifetime (see comments above). I would suggest having an estimate and quantification of your CO lifetime and budgets (e.g. like in Gaubert et al., 2016). This will help you investigating and quantifying what

is responsible for the biases in the posterior: MOPITT retrievals, LMDz or the 4DVar.

Lines 709-714: This statment now refers to MOPITT V5T, the rest of the paper is dealing with MOPITT V6J. This is confusing and probably not relevant.

Lines 713-714: The reference to the George et al., 2015 paper is misleading. It makes think the reader that is it a paper about MOPITT biases regarding IASI as a reference. This is not the goal and conclusions of George et al., 2015. Please remove, or rephrase. For a data assimilation comparison between MOPITT and IASI CO profiles, please refer to Barré et al., 2015.

Lines 721-739 and section 5.4 in general: This paragraph is not clear and to my mind drives conclusion without the necessary convincing experiments. The authors "anticipate" that assimilating CO profiles would produce larger biases. Why the authors did not assimilate the profile and then not just "anticipate" but prove this conclusion. I recommend either removing section 5.4 or provide the necessary experiments to support such conclusions.

The authors only support their conclusion by citing papers not accurately that are not using the same model and data assimilation system and experiments. For example, Gaubert al., 2016 do not infer the CO surface emissions.

Lines 749-754: The authors point out a well-known problem in chemical data assimilation. This can be overcome by using eigenvalue or more generally singular value decomposition to diagonalize R and avoid calculating off-diagonal terms of B (e.g. Migliorini et al., 2008) in a variational framework. Alternatively, approximation and ad-hoc assumption can be made on off diagonal values of B or assuming that R is diagonal and tuning diagonal values of R.

Lines 755-760: That is a shame that at the very end of the paper (and few other lines i.e. around line 665) is it stated that the sources and sinks of CO on the model could be responsible of the biases in the posterior analyses. I recommend that this should
be reinforced in the entire discussion by having further diagnostics and experiments.

Lines 760-765: I am again not sure about the validity of those statements. There is a difference between assimilating surface network data and assimilating surface re-trieved data. The representativity of this two types of data sets are fundamentally different, e.g. coverage, revisit/time-sampling, accuracy, spatial resolution. This is again very speculative, consider removing.

Line 767: "On either the satellite", syntax error, please rephrase.

Lines 776-777: Barré et al., 2015 conducted a study assimilating MOPITT and IASI and compared biases and errors with an extended set of independent observations for validation. Please refer to this paper.

References:

Barré, J., Gaubert, B., Arellano, A. F. J., Worden, H. M., Edwards, D. P., Deeter, M. N., ... Hurtmans, D. (2015).ÂăAssessing the impacts of assimilating IASI and MOPITT CO retrievals using CESM-CAM-chem and DART.ÂăJournal of Geophysical Research: Space Physics,Âă120(19), 10501-10529. DOI:Âă10.1002/2015JD023467

Inness, A., Blechschmidt, A.-M., Bouarar, I., Chabrillat, S., Crepulja, M., Engelen, R. J., Eskes, H., Flemming, J., Gaudel, A., Hendrick, F., Huijnen, V., Jones, L., Kapsom-enakis, J., Katragkou, E., Keppens, A., Langerock, B., de Mazière, M., Melas, D., Par-rington, M., Peuch, V. H., Razinger, M., Richter, A., Schultz, M. G., Suttie, M., Thouret, V., Vrekoussis, M., Wagner, A., and Zerefos, C.: Data assimilation of satelliteretrieved ozone, carbon monoxide and nitrogen dioxide with ECMWF's Composition-IFS, Atmos. Chem. Phys., 15, 5275– 5303, doi:10.5194/acp-15-5275-2015, 2015.

ÂăMiyazaki, K.,ÂăEskes, H. J., and Sudo, K.: A tropospheric chemistry reanalysis for the years 2005–2012 based on an assimilation of OMI, MLS, TES, and MOPITT satellite data,ÂăAtmos. Chem. Phys., 15, 8315-8348, doi:10.5194/acp-15-8315-2015, 2015

[Figure]

Migliorini, S., C. Piccolo, and C. Rodgers, 2008: Use of the information content in satellite measurements for an efficient interface to data assimilation. Âă Mon. Wea. Rev., Âă 136, 2633–2650.

---

## Author Comment (AC3) · 28 Feb 2018

**Reply to Reviewer 2**

Yi Yin et al.
Feb. 26 2018

Paper title: **On biases in atmospheric CO inversions assimilating MOPITT satellite retrievals**

First, we would like to thank the reviewer for his/her comments that will greatly help clarify our manuscript in the revised version. All comments are accounted for and are individually discussed hereafter.

**General Comments:**

*The submitted paper aims to evaluate the quality of CO emissions inversions and inference of other parameters influencing the CO budget using different model configurations, i.e. horizontal and vertical resolution, OH forcing and physics scheme. While the authors try to provide a detailed evaluation, and provide indications of possible biases on MOPITT retrievals and additionally of model and emissions, the methodology and scientific argumentation employed is not sound to me. It is quite unclear what are the exact goals of the paper and it seems that the focus is too broad. The evaluation is extensive which is appreciable but the discussions try to cover too many topics without robustness and convincing arguments.*

We agree with the reviewer that our discussion was a bit scattered, hence we have restructured the discussion to make our argument clearer in the revised text.

*This study needs to mature and needs to be supported by additional experiments. I fear I cannot recommend this paper for publication for the following main reasons:*

*1. The discussion and especially sections 5.3 and 5.4 are scientifically flawed. The authors drive conclusions without experimenting themselves. I strongly recommend the authors to reconsider their data assimilation experiments and setups before driving such conclusions or consider removing those two sections. I fear that without those two sections the paper will significantly loose substance. Moreover, the sensitivity tests on model parameters are not convincing, a significant increase on model horizontal resolution and using a more detailed chemical scheme would have been more useful to point out intrinsic model deficiencies and uncertainties.*

We respectfully disagree with the reviewer's harsh wording, but we agree that our articulation may not have delivered our message properly and caused some confusion. Therefore, we would like to take the chance to explain our main points, which we believe can be distilled from the experiments performed in this study and are helpful to setting the priories of future global CO inverse studies.

Section 5.3 "**Possible biases in the MOPITT retrievals**" are based on the comparison of posterior model states to other independent observations, including surface measurements, TCCON, and aircraft measurements (HIPPO/MOZAIC). Our study did not aim at evaluating the MOPITT retrievals, yet the fact that the same model state has different signs of biases compared to MOPITT and to other independent measurements reveals some inconsistency.

Section 5.4 "**Assimilating satellite total column vs. vertical profile retrievals**" are addressed "for the purpose of top-down estimates of CO emissions, in which the model cannot directly correct vertical model biases (line 769-770)" based on the vertical bias structures we showed in this study and practices of previous studies in this context:

First, we showed that (1) when updating the surface emissions the overall shape of vertical profiles can only be marginally changed (only if the profile errors stem from surface flux errors), and (2) the posterior model biases to the MOPITT profiles vary along the altitude (with opposite signs of the remaining biases between the near surface levels and the free-troposphere / stratosphere levels). Thinking logically, one would expect that when assimilating one pressure level at a time the inversion would derive different estimates of surface emissions. Indeed, Jiang et al., (2013) has demonstrated that assimilating the MOPITT surface level, the profiles, or the column amounts individually would result in different global CO emission estimates: 125.3, 150.1, and 141.9 Tg/month respectively for the period June-August 2006; this point has been further demonstrated in the follow-up studies in Jiang et al.. (2015, 2017). In such a case where model profile can only be marginally modified via the change of surface emissions, assimilating the full profiles would not bring an obvious advantage compared to assimilating the total column; instead, it poses the issue of addressing observation error correlations among different vertical levels that are difficult to characterize (note that the observation errors here are defined with respect to the inverse model, and thus, include measurement, model, and representativeness errors) and are often ignored.

Second, regarding the MOPITT retrievals, there are known temporal bias drifts in the profile retrievals, but the total column shows the strongest temporal stability as documented by Deeter et al., (2014). We noted in our paper that "the bias drift for the MOPITT V6 TIR/NIR product varies from negative in the lower troposphere ($-1.3\%yr^{-1}$ at 800 hPa) to positive in the upper troposphere ($1.6\%yr^{-1}$ at 200 hPa) as compared to aircraft measurements; yet, the bias drift in the total column is negligible (Deeter *et al*., 2014) (line 741-745)." In this case, assimilating partial profiles would bear the temporal bias drift of the partially integrated columns. In line with this expectation, Jiang et al., (2017) showed, indeed, different long-term trends in the posterior CO emissions when assimilating the MOPITT total column, the profiles, and the lower profiles (defined as the surface, 900, and 800 hPa pressure levels) (please see Figure 6 of Jiang et al., 2017). Also, correcting the latitudinal bias for each pressure level individually may degrade the overall column consistency,

which could explain partly the different results between assimilating the profile and the column, as ideally, they would result in similar budgets and trends.

Regarding reviewer's remarks on our sensitivity test, we tested model configurations that are related to the modelling of the vertical profiles: a doubling of the vertical model resolution, a different convection scheme, and a different OH field. They are intended to show weather model uncertainties associated with these aspects would impact the evaluation results particularly regarding the vertical bias structure, which is well within the scope of this study and helpful for the discussion and conclusion.

> 2.  *The quality of the scientific argumentation can be questioned. A lot of references are cited inappropriately. Number of citations do not support statements made in the present paper (see specific comments below). Demonstrations are often approximate and hand-waving. The conditional form is often used when it comes to conclusions (the forms "would" and "could" are widely used). The authors suggest and anticipate from incomplete set of experiments with few references to drive scientific conclusions.*

We thank the reviewer for pointing out some inaccuracy or confusion in our expressions. Since they are mentioned later in this review, we answer them individually below.

> 3.  *Last but not least, I am concerned about the methodology itself; statistical method- ology and the significance of the diagnostics.*

We answer this point below where more detail from the reviewer is given.

> *The reliability of the data assimilation algorithm is not discussed as well.*

This paper evaluates the prior and posterior model states with various observations including those being assimilated, which is a measure of its "reliability". The data assimilation algorithm has a 13-year history for atmospheric inversion of $CO_2$, CO, $CH_4$, HCHO, refrigerant gases and aerosols at the global scale with 10s of papers documenting its behavior or using its results. We wrote in the introduction that "this study evaluates the results of a global MOPITT CO total column assimilation using LMDz-SACS (as described by Yin *et al*. 2015) against various independent measurements for the period 2009-2011. (line 93-95)". We, therefore, assumed that we could save some space here by referring to Yin et al. (2015) and Pison et al. (2009), two papers that contain further detail and references. We have added more information in the revision.

> *To support the three mentioned points please consider the following specific comments.*

**Specific Comments:**

*Line 66: There are also other references that are using MOPITT and data assimilation to study the temporal distribution and variability of CO, e.g. Inness et al., 2015, Myazaki et al., 2015, Barré et al., 2015.*

The sentence is not about MOPITT data assimilation studies in general but about "studying the spatial-temporal distribution and variability of CO sources and sinks at the global scale" with MOPITT. Of the three references suggested here, only the second one fits the topic.

We revised the text to be more inclusive of MOPITT related studies: "The official MOPITT CO product, now reaching version 7, has played a pivotal role in the study of the spatial-temporal distribution and variability of CO sources and sinks (Shindell et al., 2006; Stein et al., 2014; Worden et al., 2013), including applications in global inverse studies (Arellano et al., 2004; Fortems-Cheiney et al., 2011, 2012; Hooghiemstra et al., 2012; Jiang et al., 2017; Kopacz et al., 2010; Yin et al., 2015) and chemical reanalyses (Barré et al., 2015; Gaubert et al., 2016; Inness et al., 2015; Miyazaki et al., 2015)."

*Line 75: Please, change plagued by another word. Models are not plagued, they just misrepresent the truth by man-made simplifications.*

We changed it to "associated with".

*Line 83-84: This is not what Hooghiemstra et al., 2012 are proving. Form the conclusion of the paper it is: "However, in the remote SH (30 − 60ÂaˇS), the comparison with MOPITT deteriorates from a 4% negative bias in the a priori to a 10% negative bias in the a posteriori solution, due to an emission decrease suggested by SH surface observations."*

We wrote "Assimilating surface in-situ measurements or MOPITT CO retrievals using the same CTM and inversion configuration could result in considerable differences in the posterior surface fluxes (Hooghiemstra *et al.*, 2012)", which, in our opinion, is a fair rephrase of the findings regarding the posterior fluxes of the Hoogiemstra et al., (2012) entitled "Comparing optimized CO emission estimates using MOPITT or NOAA surface network observations". As presented in Figure 9 and Table 1, their posterior global annual emissions assimilating station-only vs. MOPITT-only are 724±75 vs. 806±69 for anthropogenic sources, 704±78 vs. 733±60 for Natural + NMVOC, and 394±60 vs. 304±28 for biomass burning.

*Line 86: Gaubert et al., 2016 is not inverting surface emissions as Yin et al., 2015.*

We will remove this reference from the sentence.

*Lines 84-87: This statement is not well supported by either reference provided. For example, Barré et al., 2015 that is assimilating two types of sounders find opposite conclusions. MOPITT assimilation still underestimates CO at the surface over CONUS. This is probably only true in the southern hemisphere.*

Our statement is supported by Yin et al., 2015 (First four columns of Table 2, cited below), in which the mean posterior model biases compared to the surface observations are positive.

| Regions | MOPITT | | Surface | |
|---|---|---|---|---|
| | Prior bias | Post bias | Prior bias | Post bias |
| | | (ppb) | | |
| BONA | −14.1 | 0.7 | −20.8 | 20.5 |
| USA | −16.7 | −3.1 | −20.4 | 20.1 |
| NHSA | −10.0 | −3.0 | | |
| SHSA | −14.2 | 0.2 | | |
| NHAF | −15.0 | −2.0 | −20.0 | 6.8 |
| SHAF | −16.5 | 0.5 | −1.5 | 13.6 |
| WSEU | −16.1 | 0.3 | −36.7 | 18.7 |
| ESEU | −16.8 | 0.4 | | |
| BOAS | −17.3 | 1.0 | | |
| MIDE | −16.5 | −2.9 | | |
| SCAS | −12.0 | −0.3 | | |
| SEAS | −20.1 | −3.6 | −30.6 | 22.6 |
| AUST | −15.5 | −1.7 | −6.4 | 15.4 |
| INDO | −3.8 | 0.2 | | |
| OCEAN | −11.9 | −0.2 | −15.1 | 9.6 |

It is also supported by Hooghiemastra et al., (2012) (Fig. 7 copied below). The posterior (MOPITT) concentrations (in red) are in many cases higher than the ground measurements (in black), not limited to the Southern Hemisphere.

[Figure]

**Figure 7.** Prior and posterior simulation for 2004 sampled at 6 NOAA stations that were assimilated in the 4D-Var inversion stations-only. The simulation in red used optimized emissions from the MOPITT-only inversion. Black dots represent the NOAA flask observations. Error bars denote the total observation error (including the model representativeness error). The comparison with additional stations is presented in the auxiliary material.

Consistent with the previous point from the reviewer, we are considering here inverse modelling studies, which do not include Barré et al. (2015).

While it is true that Barré et al. (2015) showed "*MOPITT assimilation still underestimates CO at the surface over CONUS* (CONUS means the Continental United States using air quality data from EPA)", the authors also noted that "The suburban monitoring sites could be often located near strong localized pollution sources creating a strong source-to-receptor relationship that is a challenge to duplicate in coarse model resolution [Pfister et al., 2011].We hence **expect the models and data**

assimilation experiments to be significantly negatively biased, due to the difference of representativeness between the surface sites (local measurements) and the model having a 2×2° representative grid box."

> *Lines 87-90: This statement is not clear at all. Please clarify.*

We rephrased the sentence as "Jiang et al., (2013, 2015) further demonstrated that assimilating the MOPITT surface level (or the three near surface levels), the profiles, and the total column amounts would result in different posterior CO emission estimates, suggesting some inconsistency in the information provided to the inversion system on the vertical CO distribution between the CTM and the MOPITT data".

> *Line 88: differences between what and what?*

We mean different posterior emission budgets when assimilating the MOPITT CO profiles or when assimilating the MOPITT surface layer retrieval.

> *Line 90: While the statement is unclear to me, I do not think this is the Jiang et al., 2015 conclusions.*

Jiang et al. (2015) showed that assimilating the MOPITT CO profiles or the near surface layer retrievals results in different surface emissions. We interpret this result in terms of inconsistent information provided to the Bayesian cost function from the surface layers and from the profile data given the same control variable. We have rephrased our expressions to make it clearer.

> *Line 135: The authors should know what Bayesian means. There is nothing Bayesian in this equation.*

We respectfully disagree: the formulation actually comes from the linear Bayesian estimation framework (see, e.g., Rodgers and Connor, 2003, doi:10.1029/2002JD002299). Still, we will rephrase the sentence.

> *Line 140: Clarify the statement, it sounds as you model a profile from measurements.*

The paragraph has been revised, further to this and M. Deeter's comments.

> *Line 176: 2.5 by 3.75 degrees is now considered as low resolution, change accordingly.*

We have changed it to low resolution accordingly.

> *Line 179: is it another model? I believe you still use LMDz but with a different configuration. Change accordingly.*

We changed the sentence as "We also include a higher resolution configuration of LMDZ-SACS with a higher vertical resolution of 39 eta-pressure levels for sensitivity tests, noted as Medium Resolution (MR)."

> *Line 181: Does changing just the latitudinal resolution from 2.5 to 1.89 degrees relevant? It is then mainly a significant increase on the vertical resolution. Why not keeping the same horizontal resolution? Again 1.89 by 3.75 by 39 levels is not considered nowadays as high resolution.*

We changed the notations to medium resolution (MR) and low resolution (LR) respectively. Indeed, the major features we are interested in is the impact of the change in the vertical resolution.

> *Lines 199-200: 2009 to 2011? From what month to what month? It could be almost three years to almost one year though.*

For three full years from Jan. 2009 to Dec. 2011.

> *Lines 316-317: The authors should detail exactly how they apply the observation operator to retrieve Xmod. Have they smoothed the model profile by the averaging kernel, have they considered interpolating partial columns from the model and then convert to log(vmr) to match the MOPITT data? The authors should refer to Barré et al., 2015 section 2.2.4 for the correct approach. I am then uncertain if the method used by the author is the correct one, hence I am doubtful about the validity of the results and discussion about the MOPITT profiles validation in rest of the paper.*

-*"Have they smoothed the model profile by the averaging kernel?"*
Yes. It was noted in our manuscript that "The MOPITT-equivalent model CO vertical profiles are retrieved in a consistent manner as the satellite retrievals using Equation (2) described in section 2.1, in which the $\chi_{mod}$ become the CTM concentrations interpolated at the MOPITT pressure levels to be convolved with corresponding MOPITT prior CO profiles and averaging kernels. (line 315-318)".

-*"have they considered interpolating partial columns from the model and then convert to log(vmr) to match the MOPITT data? "*
Yes. It was noted in our manuscript that "The vector quantities $\chi$ are expressed as log10 of the corresponding volume mixing ratio (line 143-144)".

-*"The authors should refer to Barré et al., 2015 section 2.2.4 for the correct approach. "*
We strictly followed the conversion as documented in the MOPITT retrieval papers and referenced to the sample code provided in User's Guide V4 (Section 7.3 for the profile and Section 7.4 for the total column, page 18-20). We also updated the pressure grids following User's Guide V5, which applies to the V6 products as well. As this information is well documented with the data products and should be followed by default, we do not think it necessary to duplicate such information in the paper, which is not specific to our practice anyhow.

We combined this section (2.4.4) with the section about MOPITT retrieval (2.1) and described our approach as "When comparing column or profile retrievals with the model, we follow the MOPITT Version 4 User Guide (Deeter, 2009) to calculate the model-equivalent to the MOPITT retrievals, expressed in terms of logarithm of the volume mixing ratio, with the proper degree of smoothing and a priori dependence (We note that the association between retrieval levels and atmospheric layers are updated since V5) ".

> *Lines 323-324: Does this mean that you are taking the nearest grid point. If yes, is that appropriate? Since you are doing DA science you should be able to interpolate at the right location.*

Yes, for the surface observations and the aircraft measurements, we take the nearest grid point and time step where/when the observation was made because we consider that the model values are volume averages rather than point values. The link made by the reviewer between the alternative choice and the fact that we are doing DA science remains obscure to us.

> *Lines 325-328: This is unclear to me, what operation the authors are doing here. Are you shifting or scaling the profile in order to keep the same total column value? What is the "uncertainty from vertical resolution change on the CTM"? Please rewrite, develop, explain better.*

Vertical interpolation may change the mass of the profile per unit area, for instance because the surface pressure is not the same in the initial and final vertical grids. We therefore scale the profile to conserve the pressure-weighted column-mean CO concentration. Note that we exclude observations whose surface pressure differs more than 50hPa from the model state.

> *Lines 330-333: It is unclear to me what the authors are doing exactly. Are they averaging monthly model values and then they are comparing with monthly averaged observations? If yes, the entire results of this paper would be flawed. Or are they interpolating model to observation at the right time.*

The models are sampled at the time of the observation with a minimum sampling time step of 30 minutes, as described in the text "Model values are sampled according to the time and location of the measurements at a temporal resolution of 30 minutes and the model associated spatial resolution (line 321-323)". Then, they are averaged per month for each model grid for further analysis, we revised this sentence as "the observed and simulated values are averaged respectively per month for each station or model grid for further analysis (line 330)".

> *Moreover, it seems that the correlations in the rest of the paper are made on monthly averaged biases, reducing the sample for correlation to something small and probably not statistically significant. Looking at the correlations plots I see around 12 to 14 point as a sample size. Would it be statistically more sound to calculate those correlations using the entire sample of observations (not reduced by average biases)? I am then doubtful about the robustness of this score during the further analysis of this paper.*

The analysis was based on monthly averaged values. This choice was made to assess the agreement between model and observation for general spatial patterns other than fine scale variations. Averaging observations and corresponding model outputs by month also helps to weigh the observations evenly, so that the results are less impacted by uneven numbers of available observations in space and time.

If the correlation plot here refers to Fig. 3, each point represents the mean annual bias to the near surface concentration at a certain station (x-axis) and to the corresponding MOPITT $X_{CO}$ that fall within the same model grid (y-axis). Again, this plot is intended to show the spatial covariations of the model biases with respect to the surface observations and to the MOPITT $X_{CO}$.

> *Line 358: Please recall what are those big-regions. Cite Yin et al., 2015.*

We added this information in the revised manuscript in the method section. "the column-mean OH concentrations are also optimized by scaling factors over six big boxes of the atmosphere: three latitudinal boxes (30–90°S, 0-30°S, 0–30°N) and three longitudinal boxes north of 30°N (North America: 180–45°W; Europe: 45°W–60°E; Asia: 60–180°E)".

> *Line 370-376: This entire paragraph is confusing to me, please rewrite.*

We rephrased it as "The average difference of the simulated $X_{CO}$ between the posterior and the prior simulation using the reference model (LR) is ~10.4 ppb at the global scale, a much larger value than that induced by different model configurations given the same CO sources and sinks (~1.5 ppb between MR and LR) or by the two OH fields (~2.8 ppb between TransCom and INCA at the global scale, with some cancelling effect between the NH and the SH). This comparison suggests that the increment of surface emissions in the posterior estimates compared to the prior is robust with respect to uncertainties in OH distribution, vertical model resolution, and convection / boundary layer mixing scheme that we tested here."

> *Line 427: Higher sensitivity of what to what?*

We rewrite this sentence as "Larger differences of model results between MR1 and MR2 are found in the surface level [co] concentrations than the differences in the simulated $X_{CO}$, suggesting that the convection scheme has a larger impact on the boundary layer mixing ratio than the convolved total column amount".

> *Line 432-433: Please see my comment above about the significance and robustness of those correlations. The statistical methodology as it is presented now is not sound to me.*

As stated above, the correlations we presented here are spatial correlations between the annual mean model-data bias in the surface [co] at ground stations and in the total column [$X_{CO}$] at corresponding model grid. Only significant correlations with a confidence level higher than 95%

are shown (e.g. we noted "No significant correlations are found in the posterior simulations over the NH (**Figure 3b**), suggesting no consistent systematic errors of the MOPITT $X_{CO}$ assimilation compared to surface [CO] measurements").

> *Line 443-445: This is again confusing with a "hand-waving" argument displaying only the HR correlations, using the word "likely" and not further investigating the possible error on the vertical error CO profile on the posterior. Additionally, I would not trust a correlation of 0.49 with a sample size of 14 using monthly averaged biases.*

The sample size is the number of available stations. This correlation is statistically significant as stated above. The choice was made to show spatial covariations.

> *Line 469: The word contamination is not appropriately used. For example, there is contamination in data when an instrument is not working correctly and generate a systematic error. Please replace this word.*

We have revised the sentence as "This could be partly explained by strong local emissions in the boundary layer that are measured by MOZAIC observations near airports, but are not represented by our global model".

> *Lines 474-479, lines 506-513, figures 4 and 5: MOZAIC profiles are most likely close to the sources whereas HIPPO measures remote scenes. The bias observed in the posterior for HIPPO profiles are due to an overly long CO lifetime in the simplified chemistry model. The presented data assimilation system infers the surface CO sources but do not directly corrects for CO lifetime error due to an (over) simplified chemical scheme.*

In our system, the average CO lifetime is constrained by the MCF loss rate. However, the reviewer's explanation does not contradict our text "Such bias in representing the oceanic vertical profiles suggests error in the vertical distribution of CO source/sink over ocean or in the vertical mixing. "

> *Lines 531-532: Please rephrase. "The MOPITT profiles are well reproduced by the model. . .". The model does not reproduce MOPITT profiles.*

The entire sentence is "The MOPITT vertical profiles are well reproduced by the model north of 30°N, but considerable deviations are found for the vertical structure over the tropics and subtropics", which we think is a fair description of Fig. 6d for the posterior model results (dash-dot blue line).

> *Lines 541-544: In Deeter et al., 2014, the MOPITT V6 validation with HIPPO do not see such errors in the upper troposphere. Also, the author should also take into account the spatial sampling of MOZAIC, HIPPO versus MOPITT. The longitudinal distribution of CO, in the tropics can be highly variable.*

Actually, in Deeter et al., (2014), for the years we study here 2009-2011, there are some significant positive biases when compared to NOAA profiles, as shown below on the left, cited from **Fig. 9** of Deeter et al., (2014). The authors also showed the latitudinal distribution of biases with respect to HIPPO for the TIR-only retrieval at 200 hPa, where you can see positive biases in the tropics (as shown below on the right, cited from **Fig. 10** of Deeter et al., (2014)).

[Figure]

It is documented in the text that "V5 TIR-only validation results exhibited a strong latitude dependence in the retrieval bias at 200hPa, with biases of about 25% in the tropics. In contrast, the maximum retrieval bias in log(VMR) at 200 hPa in the V6 results is less than 0.04, which is equivalent to about 10%. V6 results for other retrieval levels are similar to V5 results, with log(VMR) biases at all levels and all latitudes generally less than 0.05." Our analysis (based on V6J) are thus in line with the results of Deeter et al., (2014), but presented in different quantities using a different manner.

Line 565: *Which tropical ocean? Rephrase.*

We did not look into longitudinal variations. The analysis is for the 30S-30N ocean.

Line 574: *"over the ocean"*

Corrected

Line 607: *What are those big regions? Recall or cite Yin et al., 2015.*

We added this information in the methods. Please see reply to Line 358.

*Lines 638-643: What is the purpose of this paragraph? It is not clear what the authors are trying to demonstrate. Please clarify, develop, rephrase.*

The purpose of this paragraph is to show that "The differences in surface [co] concentration and in the total column $X_{CO}$ at the corresponding model grid are not correlated when using different model configurations (MR vs. LR or MR2 vs. MR1), which suggests that there is not a systematic impact on both the surface [CO] and the $X_{CO}$ comparing one model configuration to another. In other words, it shows that if one model produces a higher $X_{CO}$, it does not necessarily produce a higher surface [CO] at the same time. As MR consistently produces a slightly higher $X_{CO}$ compared

to LR, we could deduce that assimilating the same MOPITT $X_{CO}$ observation using MR would derive a smaller CO emission estimates compared to current LR inverse results."

*Lines 649-652: The syntax of this sentence is not correct.*

We rephrased the sentence as "The negative prior simulation biases in the surface [co] and in the Xco are also highly correlated in space (Fig. 3a)".

*Lines 683-695 and section 5.3 in general: The conclusion of "positive biases in the MOPITT retrievals" is flawed here. The authors utilize only one inversion technique from only using total column product. They infer only the surface emissions that is not a direct measured quantity from MOPITT CO retrievals. Depending of a model quality (i.e. resolution, chemistry, horizontal and vertical transport, and so on. . .) inverting the emissions only can lead to good result for the wrong reasons and conversely often having the "correct" emissions and having significant errors in the atmosphere. Data assimilation rely on observation but ALSO on models, you could have the best observation quality, if the model is inaccurate the analysis and the subsequent forecasts would be degraded. Before jumping quick in such important conclusions several things should be tested carefully such as:*
*Assimilating the CO fields directly with total CO columns and CO profiles*
*Rerunning the current experiments with a more complete and detailed chemistry*

Our remark of "positive biases in the MOPITT retrievals for the total column and at the near surface levels" are made specifically for regions "in the mid- and high latitudes", where the posterior biases are consistent comparing to both the surface and TCCON observations. The reviewer's argument does not explain the residual bias to TCCON. Further, a similar bias structure is documented by Jiang et al. (2017). We have added more discussion regarding previous studies documenting MOPITT bias information (e.g. Deeter et al., 2014, Gauber et al., 2016) in the revised manuscript to enrich the discussion.

*Line 692-695: Deeter et al., 2014 made the direct comparisons between MOPITT V6T (which is the same as MOPITT V6J over the ocean) and HIPPO measurements: providing a quantification of the MOPITT biases: 1.5% 7.7% at the 200hPa level. How can the authors can explain such discrepancies with those results and figure 4 and 7. The authors compare figure 2 and 4 with figure 7. Again, the representativity of the statistics made here should be considered. HIPPO and MOZAIC cover specific regions whereas MOPITT provide a global picture. Is it reasonable to compare those figures in order to drive conclusions about biases without quantification?*

Figure 6 of Deeter et al. 2014 shows +1.5% bias (MOPITT too high) and 7.7% std at 200 hPa for MOPITT vs. HIPPO misfits in terms of log(VMR). This is not inconsistent with what we show in Fig.

7, with zonally-averaged model-MOPITT misfits expressed in terms of VMR about -20 ppb around 200 hPa.

*Lines 702-707: This indicate an issue in your CO lifetime (see comments above). I would suggest having an estimate and quantification of your CO lifetime and budgets (e.g. like in Gaubert et al., 2016). This will help you investigating and quantifying what is responsible for the biases in the posterior: MOPITT retrievals, LMDz or the 4DVar.*

The CO budgets were analyzed in Yin et al., (2015) in Fig. 8. We added discussion regarding this point in the revised text. Note that the posterior model states agree well with the MOPITT total column as shown in Fig. 2 for the average model-data bias along the latitude, Fig. 8 for the regional statistics, and Table 3 for the zonal statistics.

*Lines 709-714: This statment now refers to MOPITT V5T, the rest of the paper is dealing with MOPITT V6J. This is confusing and probably not relevant.*

We also described in the text that "The differences between these two instruments (ranging from 0-13%) are significantly larger than that between MOPITT v5 and v6 [Deeter et al., 2014]."

*Lines 713-714: The reference to the George et al., 2015 paper is misleading. It makes think the reader that is it a paper about MOPITT biases regarding IASI as a reference. This is not the goal and conclusions of George et al., 2015. Please remove, or rephrase. For a data assimilation comparison between MOPITT and IASI CO profiles, please refer to Barré et al., 2015.*

We cited information that is relevant to our discussion from the study of George et al., 2015, which focused on the comparison between MOPITT and IASI CO column, including the influence of their associate prior profile. We changed the word "biases" in line 712 to "differences". We also added references to Barre et al., 2015.

*Lines 721-739 and section 5.4 in general: This paragraph is not clear and to my mind drives conclusion without the necessary convincing experiments. The authors "anticipate" that assimilating CO profiles would produce larger biases. Why the authors did not assimilate the profile and then not just "anticipate" but prove this conclusion. I recommend either removing section 5.4 or provide the necessary experiments to support such conclusions.*
*The authors only support their conclusion by citing papers not accurately that are not using the same model and data assimilation system and experiments. For example, Gaubert al., 2016 do not infer the CO surface emissions.*

We showed that 1) updating the surface CO emission would not change significantly the shape of the vertical profile, 2) after fitting the total column there are still negative simulation biases compared to the near surface level MOPITT retrievals. Therefore, it is reasonable to deduce (anticipate) that assimilating those profile retrievals of near surface levels (from surface up to 700

hPa) would derive a larger surface emissions and thus even larger positive biases compared to ground and TCCON observations.

> *Lines 749-754: The authors point out a well-known problem in chemical data assimilation. This can be overcome by using eigenvalue or more generally singular value decomposition to diagonalize R and avoid calculating off-diagonal terms of B (e.g. Migliorini et al., 2008) in a variational framework. Alternatively, approximation and ad-hoc assumption can be made on off diagonal values of B or assuming that R is diagonal and tuning diagonal values of R.*

Our point is not about the technicality of assimilating profiles (apart from the last line where we briefly acknowledge the fact that most inversion systems do not handle full error covariance matrices). Rather, our point is about the fact that we have hardly any basis to guide the assignment of the error covariance matrix from the model equations.

> *Lines 755-760: That is a shame that at the very end of the paper (and few other lines i.e. around line 665) is it stated that the sources and sinks of CO on the model could be responsible of the biases in the posterior analyses. I recommend that this should be reinforced in the entire discussion by having further diagnostics and experiments.*

We invite the reviewer to read our paper again. In many places, model errors are discussed. In the abstract, we wrote "Biases in representing vertical CO profiles are found over the ocean and most significantly in the Southern hemisphere, suggesting errors in the vertical distribution of CO chemical sources/sinks or in the vertical mixing to be improved in future modelling studies"; then, we described in the introduction that "model results are still associated with systematic uncertainties due to a complex interplay of source distributions, transport and chemistry" and "In order to diagnose these inconsistencies that could result from model errors or satellite retrieval biases or both". Other places like lines 443-445, 470, 511-513, 599-666, 669-680 all discussed model errors. We will better organize the discussion to reflect this point.

> *Lines 760-765: I am again not sure about the validity of those statements. There is a difference between assimilating surface network data and assimilating surface re- trieved data. The representativity of this two types of data sets are fundamentally dif- ferent, e.g. coverage, revisit/time-sampling, accuracy, spatial resolution. This is again very speculative, consider removing.*

Our statement is actually a verbatim from (Rayner and O'Brien 2001). We also referred to "Because there is so much less spatial variability associated with column $CO_2$, even with just a few observations, it will be possible to assess the strength of the Northern Hemisphere carbon sink using a measurement approach that is far less sensitive to model representations of vertical mixing" (Olsen and Randerson, 2004) and to "Because these observations are of the column or partial column abundance, they come close to directly representing a measure of atmospheric $CO_2$ mass per unit area. As a result, our estimate of NEE are less sensitive to errors in the vertical transport than estimates based solely on surface mixing ratio observations" (Yang et al. 2007). None of these

sentences from past papers, either explicitly or implicitly referred to coverage, revisit/time-sampling, accuracy, or spatial resolution.

> *Line 767: "On either the satellite", syntax error, please rephrase.*

We have replaced "either … or" by "both … and".

> *Lines 776-777: Barré et al., 2015 conducted a study assimilating MOPITT and IASI and compared biases and errors with an extended set of independent observations for validation. Please refer to this paper.*

We added a reference to Barre et al. (2015).

> *References (from the reviewer):*
> *Barré, J., Gaubert, B., Arellano, A. F. J., Worden, H. M., Edwards, D. P., Deeter, M. N., ... Hurtmans, D. (2015). Âa ˘Assessing the impacts of assimilating IASI and MOPITT CO retrievals using CESM-CAM-chem and DART.Âa ˘Journal of Geophysical Research: Space Physics,Âa ˘120(19), 10501-10529. DOI:Âa ˘10.1002/2015JD023467*
>
> *Inness, A., Blechschmidt, A.-M., Bouarar, I., Chabrillat, S., Crepulja, M., Engelen, R. J., Eskes, H., Flemming, J., Gaudel, A., Hendrick, F., Huijnen, V., Jones, L., Kapsom-enakis, J., Katragkou, E., Keppens, A., Langerock, B., de Mazière, M., Melas, D., Par-rington, M., Peuch, V. H., Razinger, M., Richter, A., Schultz, M. G., Suttie, M., Thouret, V., Vrekoussis, M., Wagner, A., and Zerefos, C.: Data assimilation of satelliteretrieved ozone, carbon monoxide and nitrogen dioxide with ECMWF's Composition-IFS, Atmos. Chem. Phys., 15, 5275– 5303, doi:10.5194/acp-15-5275-2015, 2015.*
>
> *Âa ˘Miyazaki, K.,Âa ˘Eskes, H. J., and Sudo, K.: A tropospheric chemistry reanalysis for the years 2005–2012 based on an assimilation of OMI, MLS, TES, and MOPITT satellite data,Âa ˘Atmos. Chem. Phys., 15, 8315-8348, doi:10.5194/acp-15-8315-2015, 2015 Migliorini, S., C. Piccolo, and C. Rodgers, 2008: Use of the information content in satellite measurements for an efficient interface to data assimilation.Âa ˘Mon. Wea. Rev.,Âa ˘136, 2633–2650.*

We did not include a reference list here, since they have been included in the original manuscript or introduced by the reviewer.

---

## Author Comment (AC4) · 28 Feb 2018

**Reply to Reviewer 1. B. Gaubert**

Yi Yin et al.
Feb. 26 2018

Paper title: **On biases in atmospheric CO inversions assimilating MOPITT satellite retrievals**

First of all, we thank the reviewer for his interest in our work and for his thorough and constructive review. Please see detailed replies below to each point being raised.

**General Comments:**

*This paper aims to evaluate the representation of the CO fields obtained from the assimilation of MOPITTv6 total column (XCO) retrievals (one experiment) and several modelling sensitivity tests. The set of parameters from those sensitivity tests includes the use of posterior emission fields, OH fields, and transport through different grid spacing's, boundary layers and convection schemes.*

*Further research is needed in some studies using data assimilation to investigate, quantify and in fine understand biases from satellite retrievals, model simulation, and potentially from the assimilation methods (observation operator, error characterization, state vector choices). The comparison of analysis/posterior products can provide a different perspective for the nature of those biases, different from instrument validation and model evaluation by itself. In addition, the use of different and independent datasets is usually informative about the quality of the assimilated and/or posterior model fields. However, the paper's objectives, methods and conclusions need stronger clarifications. In other words, the conclusions can be misleading with regards to the scientific methods used. The authors should either review their conclusions or consider another evaluation approach before publication.*

As suggested, and as explained in the following, we have carefully revised our text to avoid confusion. We believe that the revised manuscript better delivers our intended message.

*From the conclusions/abstracts, it is stated that the "purpose of top-down estimates of CO emissions, in which the model cannot directly correct vertical model biases, it is more robust to assimilate the column than a particular pressure level retrieval, a partial profile [. . .]".*

*As it is known, assimilating total columns presents some advantages. For instance, total columns can have lower instrument biases. For pragmatic reasons, it makes sense and I agree that if the observation does not give any information about the*

*vertical distribution, it is more appropriate to not correct for the vertical distribution within the assimilation scheme. However, those conclusions are surprising with regards to the actual work that has been done in the paper. There are no tests that would rigorously test the hypothesis of choosing a method versus another. In addition, some comments are contradictory.*

Our argument is not only based on the measurement features that the reviewer highlights here. In data assimilation / atmospheric inverse modelling, the observation errors are defined with respect to the model, and therefore, combine measurement, model, and representativeness errors. Our paper is focused on model errors and suggests large model flaws in the vertical distribution of tracers that need to be accounted for in the data assimilation, but that are not, or cannot be, well characterized in the form of an error covariance matrix.

*One important point to consider is that assimilating total columns relies on the vertical profile given by the model, thus, it is just the total column abundance that is shifted. Within that framework, there is no reason to expect improvement of the vertical profile, because it means that the modelled vertical profile is assumed to be perfect (relatively to the satellite observation). Or it can be assumed that it is far to be the case in global models (with coarse resolution in both vertical and horizontal). It is also acknowledged in the paper: "From the CTM perspective, the evaluation against aircraft measurements reveals significant model errors in representing the vertical CO gradient, in particular over the ocean."*

Atmospheric inverse modelling can improve the modelled vertical profile if the profile errors stem from surface flux errors. Indeed, model equations are left unchanged while model simulations are adjusted through their boundary conditions. Our point is that the model equation errors are so large that their statistics have to be accounted for in the inversion (both weak- and strong-constraint systems), even though they are not well characterized.

*Moreover, those errors persist after optimizing emissions, which means that chemistry and transport are both important. This is also acknowledged in the paper, but in the abstract, it is mentioned that: "Consistent negative prior biases to all types of observations in all sensitivity tests suggest an underestimation of current surface emissions in the Northern hemisphere. In contrast, prior simulations fit the surface air sample observations well in the Southern hemisphere but underestimate CO in the free troposphere and on average in the column."*

*For instance, Stein et al. (2014) demonstrated that the northern hemisphere spring cannot be attributed to direct CO emission alone. Myazaki et al. (2015) with a simple sensitivity test, a change in the CO + OH reaction rate were able to considerably reduce this bias. These studies suggest that this bias is likely to occur due to a combination of errors from chemistry, deposition, direct and indirect emission processes (vertical distribution, time profile), as well as transport. Far from the sources (in the free troposphere, over the ocean, in the southern hemisphere), it is even more likely that the problem will be due to transport and*

*chemistry (secondary sources of CO and/or the OH sink). In particular, biases can be shared through the CH4/CO/OH system, as Strode et al. (2015) and Elshorbany et al. (2016) posit. In general, please consider discussing those previously published results with regards to the setup used in this study, which presents some advantages, but also some drawbacks to compare and contrast different possible approaches.*

We thank the reviewer for pointing this out. We over-simply summarized the contrast between the NH and SH results in the abstract. Indeed, both emission estimates and CTM errors have contributions to the model-data mismatch. It has been revised as "In the Northern hemisphere, we find consistent negative biases in the prior simulations when compared to surface, aircraft, and satellite data; in contrast, prior simulations fit the surface observations well in the Southern hemisphere but underestimate CO in the free troposphere and on average in the total column."

We also note that the reviewer reinforces our point about the importance of model errors, that may already appear when assimilating the retrieved total column rather than the retrieved profile. We have added more discussion with respect to previous publications in the text when discussing the bias patterns.

*In order to improve the study please consider the following:*

*1. If this paper is aimed to evaluate the impact of assimilating either the profile or the total columns, this can be done using data assimilation experiments together with observation space diagnostics, see El Amraoui et al. (2014). The use of innovation statistics (and data assimilation) diagnostics allows to quantify the bias, while taking into account error variances from both model and observations. In particular, looking at those by Assimilation of total columns and evaluation of profiles (in observation space) or Assimilation of profiles and evaluation of columns (in observation space).*

*By doing this the diagnostics for the assimilated and non-assimilated observations can be run for each case to verify the consistency between the columns and the profile in the observation space. In a case where assimilation parameters are correct (with regards to those same diagnostics) and underlying assumptions are not too violated, it would give a more robust estimate of the impact on the total columns while assimilating profiles and vice-versa.*

Our paper actually shows one side of the diagnostics suggested by the reviewer (assimilation of total columns and evaluation of profiles in observation space). The other side implies assimilating the profiles and therefore assigning error statistics for the model capability to distribute CO in the vertical. Such a choice of a covariance matrix would drive the inversion results to a large extent but would be mostly arbitrary. The situation is different for the study of El Amraoui et al. (2014) that controlled the 3-D CO field directly rather than the emissions. The study of El Amraoui et al. (2014) also concluded that, "for this kind data (MOPITT V3)"—referring to dataset with relatively low degree of freedom for signal (DFS, ~1.5 for vertical profiles and ~1 for the total column)—

"the present method consisting of deducing the profiles from the total columns remains valid when only using the adjoint of the integration operator".

> *2. Other studies that aim to improve the model error representation in chemical data assimilation (e.g., Gaubert et al. 2016; Emili et al. 2016 and reference therein) as well as potentials from strong constraint 4D-Var (e.g. Trémolet 2006) could be discussed and analyzed.*

Gaubert et al. (2016) highlight for model-error accounting as a future direction from their work and we certainly agree. Weak-constraint approaches as in Trémolet (2006) or Emili et al. (2016) represent interesting possibilities for this, but will only be efficient if there is enough information content in the assimilated data to disentangle emission errors from model equation errors. As of today, the response to this question is unclear and necessitates much research.

> *3. The assimilation of compact phase space retrievals (CPSRs) could be considered, which is an alternative approach to profile assimilation (e.g. Mizzi et al. 2016).*

We understand that this suggestion answers our sentence "measurement error correlations are commonly ignored in inversion systems for technical reasons". Mizzi et al. (2016) presented an interesting approach to assimilate profile retrievals with coarse vertical resolution and is referred to in the revised version. It does not solve the problem of assigning model error statistics.

> *4. Thanks to the comparison to HIPPO measurements, the first identification of an upper tropospheric bias last from the MOPITT V4 (Deeter et al. 2010). The identification of the bias and update of the statistics against HIPPO has continued since then (Deeter et al., 2013; Deeter et al., 2017). You can review Martínez-Alonso et al. (2014) for another evaluation of MOPITT profiles and satellite data. Jiang et al. (2013) suggest not to assimilate the profiles in the upper troposphere while Jiang et al. (2017) propose a latitudinal bias correction. There is evidence that errors can arise from the multispectral retrieval for the nighttime oceanic scene (Worden et al., 2010).*

We have included most of the above-mentioned studies and findings in our original manuscript. We thank the reviewer for bringing up the ones that were omitted. We added more discussion regarding the comparison to previous studies in the revision.

> *5. Finally, please consider to include an extensive discussion on the sources of model error and limitation of the setup of the simulations. You can provide more quantitative results. To do so, it is highly recommendable to use conservative measures to compare simulations such as atmospheric burden for CO abundance and RMSE (or absolute errors) for comparison of simulation with observations. On the spatial resolution, one can also discuss the fact that emissions estimates would be different at higher resolution. The change of resolution changes the distribution of emissions and thus chemical regimes. The errors evolution with*

*regards to spatial grid spacing can be not linear in chemical transport models. Here, the emissions have been optimized at lower resolution, this leads to serious limitation to conclude about the effect of spatial resolution.*

We thank the reviewer for making this suggestion. We extended the discussion much following the advice here and the ones above. Also, following the reviewer's suggestions, we added information on RMSE for comparison of simulation with observations (Fig. S1, please see below); we kept the ones estimating mean biases in the main text, given that the sign of the biases contains useful information.

Figure S1. Annual mean RMSE of the prior and posterior simulations sorted along the latitude. **(a)** and **(b)** show model RMSE compared to MOPITT $X_{CO}$ using INCA or TransCom OH field respectively. Dashed lines represent the prior simulations, whereas solid lines the posterior simulation; color codes represent model versions as noted in the figure legend. **(c)** and **(d)** show model RMSE compared to independent TCCON $X_{CO}$ measurements. Colors represent model versions with triangles denoting prior simulations and dots denoting posterior simulations. **(e)** and **(f)** show model biases compared to independent ground in-situ measurements from the surface network.

[Figure]

Regarding the setup of spatial resolution, the main point of using the higher resolution is primarily to test its impact on the vertical profiles from 19 levels to 39 levels; the change in the horizontal

resolution is marginal, remaining the same longitudinally and changing from 2.5° to 1.89° latitudinally. We have made this point clearer in the revised text.

> *6. Perhaps, a further discussion on the limitation of the methods is needed, knowing that emission is not the only error source. For instance, there is dry deposition (Stein et al. 2014), time evolution coupling and feedbacks of chemistry (Strode et al., 2015; Gaubert et al. 2016; Elshorbany et al., 2016) and vertical transport, the aggregation of VOC's oxidation in one term (e.g. Jiang et al., 2015 and reference therein), etc.*

We thank the reviewer for making these suggestions. We addressed those points at different parts of the initial manuscript but did not synthesize those into a single section, e.g. we mentioned that "Stein et al., (2014) showed that a higher winter traffic emissions from North America and Europe and a lower dry deposition rate could improve the agreement between simulated and observed CO during winter and spring. (line 652-655)" when discussing possible biases in the prior CO sources in the Northern Hemisphere; we also mentioned that "This could be partly caused by errors in the vertical distribution of tropospheric secondary chemical CO sources, as the surface emissions are generally low but important sources are from chemical oxidation of hydrocarbons and from long-distance transport (Zeng et al., 2015) (line 660-663)" when discussing the Southern Hemisphere model biases. Those points are now better organized in the discussion.

*References*

*Barré, J., et al. (2015), Assessing the impacts of assimilating IASI and MOPITT CO retrievals using CESM-CAM-chem and DART, J. Geophys. Res. Atmos., 120, 10,501–10,529, doi:10.1002/2015JD023467.*

*Deeter, M. N., et al. (2010), The MOPITT version 4 CO product: Algorithm enhancements, validation, and long-term stability, J. Geophys. Res., 115, D07306, doi:10.1029/2009JD013005.*

*Deeter, M. N., S. Martínez-Alonso, D. P. Edwards, L. K. Emmons, J. C. Gille, H. M. Worden, J. V. Pittman, B. C. Daube, and S. C. Wofsy (2013), Validation of MOPITT Version 5 thermal-infrared, near-infrared, and multispectral carbon monoxide profile retrievals for 2000–2011, J. Geophys. Res. Atmos., 118, 6710–6725, doi:10.1002/jgrd.50272.*

*Deeter, M. N., Edwards, D. P., Francis, G. L., Gille, J. C., Martinez-Alonso, S., Worden, H. M., and C. Sweeney (2017), A Climate-scale Satellite Record for Carbon Monoxide: The MOPITT Version 7 Product, Atmos. Meas. Tech. Discuss., doi:10.5194/amt-2017- 71, in review.*

*El Amraoui, L., Attié, J.-L., Ricaud, P., Lahoz, W. A., Piacentini, A., Peuch, V.-H., Warner, J. X., Abida, R., Barré, J., and R. Zbinden (2014) Tropospheric CO vertical profiles deduced from total columns using data assimilation: methodology and validation, Atmos. Meas. Tech., 7, 3035-3057, doi:10.5194/amt-7-3035-2014.*

*Elshorbany, Y. F., Duncan, B. N., Strode, S. A., Wang, J. S., and Kouatchou, J.: The description and validation of the computationally Efficient CH4–CO–OH (EC-COHv1.01) chemistry module for 3-D model applications, Geosci. Model Dev., 9, 799-822, doi:10.5194/gmd-9-799-2016, 2016.*

*Fisher, J. A., Wilson, S. R., Zeng, G., Williams, J. E., Emmons, L. K., Langenfelds, R. L., Krummel, P. B., and Steele, L. P.: Seasonal changes in the tropospheric car- bon monoxide profile over the remote Southern Hemisphere evaluated using multi- model simulations and aircraft observations, Atmos. Chem. Phys., 15, 3217-3239, doi:10.5194/acp-15-3217-2015, 2015.*

*Jiang, Z., D. B. A. Jones, H. M. Worden, M. N. Deeter, D. K. Henze, J. Worden, K. W. Bowman, C. A. M. Brenninkmeijer, and T. J. Schuck (2013), Impact of model errors in convective transport on CO source estimates inferred from MOPITT CO retrievals, J. Geophys. Res. Atmos., 118, 2073–2083, doi:10.1002/jgrd.50216.*

*Jiang, Z., Worden, J. R., Worden, H., Deeter, M., Jones, D. B. A., Arellano, A. F., and Henze, D. K. (2017), A 15-year record of CO emissions constrained by MOPITT CO observations, Atmos. Chem. Phys., 17, 4565-4583, doi:10.5194/acp-17-4565-2017.*

*Mizzi, A. P., ArellanoÂa Jr., A. F., Edwards, D. P., Anderson, J. L., and Pfister, G. G.: Assimilating compact phase space retrievals of atmospheric composition with WRF-Chem/DART: a regional chemical transport/ensemble Kalman filter data assimilation system, Geosci. Model Dev., 9, 965-978, doi:10.5194/gmd-9-965-2016, 2016.*

*Strode, S. A., B. N. Duncan, E. A. Yegorova, J. Kouatchou, J. R. Ziemke, and A. R. Douglass (2015), Implications of carbon monoxide bias for methane lifetime and atmospheric composition in chemistry climate models, Atmos. Chem. Phys., 15, 11,789–11,805, doi:10.5194/acp-15-11789-2015.*

*Trémolet, Y.: Accounting for an imperfect model in 4D-Var, Q. J. Roy. Meteor. Soc., 132, 2483–2504, doi:10.1256/qj.05.224, 2006.*

*Worden, H. M., Deeter, M. N., Edwards, D. P., Gille, J. C., Drummond, J. R., and Nédélec, P. (2010), Observations of near-surface carbon monoxide from space using*

*MOPITT multispectral retrievals, J. Geophys. Res., 115, D18 314,*
*doi:10.1029/2010JD014242, http://doi.wiley.com/10. 1029/2010JD014242.*

> *There are some sentences starting with "It seems" followed by a strong fact statement: "The choice of the prior OH seems to produce the largest differences in the simulated CO concentrations at a global scale." "representing the vertical profile correctly seems to be a grand challenge." This is confusing, please choose between hypothesis or fact.*

We have rephrased this expression as "the choice of the prior OH field produces the largest differences in the simulated CO concentration at a global scale across the sensitivity tests we performed in this study", "correctly representing the vertical profile is challenging for either the satellite retrieval or the CTM". We have made similar changes in the rest of the text.

> *The Section "2.1 MOPITT satellite retrievals of CO total column and vertical profiles" is not clear and contains errors. It needs major revision. Can you merge with section 2.4.4? I don't think the authors should presents the retrieval algorithms. Please provide instead the way they applied the averaging kernels for the columns (equation 1) and the profile (equation 2).*

We have revised this section thoroughly. The separation of section 2.4.4 from section 2.1 is made to clarify that only the total columns are used for the inversion (2.1), while the profile is used for evaluation (2.4). We have changed the structure following the reviewer's suggestion and removed the parts resenting the retrieval algorithms. The application of the averaging kernels for the columns and the profile strictly follows those two equations, in which $x_{mod}$ (used here) was modified from $x_{true}$ (used for the MOPITT convention, e.g. see Deeter et al., (2014)) to reflect that the CTM values are used for the corresponding term.

> *From Yin et al. 2015, it is stated that the MOPITT observations are average on the coarser grid. For a fair comparison, the evaluation should be done at 'higher' resolution, note that high resolution for a simulation at 1.895 by 2.5 degrees is misleading. A comparison of the impact of the vertical grid spacing should be done first, what is the observation error differences between low (2.5°x3.75°) and medium (1.895 by 2.5 degrees)? What is the differences in biases?*

Indeed, as stated in Yin et al., 2015, all observations located in the same model grid within every 30-minute time steps are averaged as a "super observation" for the assimilation; similarly, the observations are averaged at corresponding model resolutions for the evaluation, not at a fixed coarse resolution. We acknowledge that the expression 'high' resolution is a bit misleading; yet, in the community of global inverse studies, this resolution is relatively high, which was the reason

for the initial choice. To avoid confusion, in the revised text the terms are changed to low resolution (LR) and medium resolution (MR) following the comments.

Concerning the change in the horizontal grid, we do think that it is modest enough (unchanged in longitude, refined from 2.5° to 1.9° in latitude – note that the reviewer's numbers here are incorrect) not to bear significant impact on the results. We have toned down our expression "finer horizontal resolution" to avoid confusion.

**Minor Comments**

*Abstract*
*Page 1, Line 16: "Carbon monoxide (CO) inverse modelling studies have so far reported significant discrepancies between model concentrations optimized with . . . (MO- PITT) satellite retrievals and surface in-situ measurements."*
*In my opinion, this first sentence is misleading and is not fair to MOPITT itself. It is common in atmospheric composition that there are discrepancies satellite and surface observations, or between satellite themselves (Kopacz et al. 2010). It usually leads to large improvements in particular for the well-known northern hemisphere spring bias. Which means that the errors are also due to model and coarse resolution. There have been issues in the Southern Hemisphere, but the reasons are not clear. The model errors appear to not be driven by CO emission alone (e.g. Fisher et al., 2015).*

The sentence cited here stated the fact that there are discrepancies between MOPITT optimized concentrations and surface observations (as reported by previous studies), but by no means did it imply that this error comes from MOPITT. The reviewer also mentioned here that "*It is common in atmospheric composition that there are discrepancies satellite and surface observations, or between satellite themselves (Kopacz et al. 2010)*". Thus, it stated the motivation of this study to look more closely into the model-data mismatches.

*1 Introduction*
*P2, L76: "with most CTMs showing negative biases to surface and satellite observations in the Northern hemisphere when prescribed with current emission inventories (Naik et al., 2013; Patra et al., 2011; Shindell et al., 2006)." The Patra et al.'s study is about TransCom-CH4 and the paragraph is CTM's biases in Northern Hemisphere spring of CO modelling, please change this reference to Stein et al. (2014).*

We referred to Patra et al., 2011 for their study of interhemispheric transport modelling in general, but since we do not elaborate on this further we have removed it.

*P2, L85: "MOPITT-based atmospheric inversions were also shown to be biased high when compared to independent in-situ surface observations in the boundary layer (Gaubert et al., 2016; Yin et al., 2015)." Gaubert et al. (2016) showed that assimilating MOPITT improve the CO values at the surface (not the opposite). The cross-validation with FTS in the southern hemisphere suggests that the model has*

*actually a good prior for wrong reasons, the assimilation improve correlation and suggest an underestimation of biomass burning emissions. Please be more precise, at least indicate that it is in the southern extra-tropical region.*

The cited sentence did not mean that assimilating MOPITT degraded its comparison to surface observations, but that the MOPITT reanalysis was biased high when compared to independent surface observations as illustrated in Fig. 5 in Gaubert et al., (2016) (cited below), which is fair in our opinion. It is noted by the authors that "The increase of CO in MOPITT Reanalysis lead to an overestimation with respect to the observed values of around 30 ppb in the SH and less than 10 ppb in the NH". But we agree with the reviewer that more detail regarding these findings could be given in the discussion, which is now added in the revised text.

[Figure]

**Figure 5.** Annual CO average of surface WDCGG observations (black), MOPITT Reanalysis (red), and the Control Run (blue).

*P3, L107: "In section 3, we first evaluate the model with MOPITT XCO that is used for assimilation, then [. . .]". Please clarify, only one simulation is done with assimilation of MOPITT XCO, what is the model?*

The details regarding the set-ups are given later in section 2. We kept the model description simple in the introduction of the paper structure where different versions are not mentioned yet at this point. Some hints were given in the previous paragraph that "this study evaluates the results of a global MOPITT CO total column assimilation using LMDz-SACS (as described by Yin *et al*. 2015) (line 93-94)", and "We also include a series of sensitivity tests (described in Section 2.2 and 2.3) to discuss model uncertainties (line 101-102)". We have revised the manuscript extensively to better describe our simulations.

*P4 L152: change "coverting" to converting*
Changed.

*P4, L186: "HR2 corresponds to the version called standard physics (SP) in (Locatelli et al., 2015), where more detail regarding these configurations are described." Please rephrase, e.g. "The latter corresponds to a version called*

*standard physics (SP), presented in Locatelli et al. (2015), where more details about these configurations are described."*

Thanks. It is implemented as suggested.

*P5, L189: Please rephrase, "as suggested by Radon [. . .]", to "as suggested by the comparison with Radon [. . .]".*

This has been corrected.

*P5, L191: Please rephrase "For all versions of LMDz-SACS here", For all the different LMDz-SACS configurations presented in this study: [. . .] boundary conditions and horizontal winds [. . .]"*

Done.

*P5: L214: "One scale factor is applied to the HR results over the globe to conserve the global mass budget to be consistent with the reference version MR." Can you explain and be more explicit? If the 3D fields of Formaldehyde have been simulated using different spatial resolution, why would not you keep a different field? Which budget (CO or HCHO)?*

The integrated global burdens of the 3D $CH_2O$ fields produced by the full chemistry model with MR and HR resolutions are slightly different. The former was optimized during the assimilation together with MOPITT $X_{CO}$. We keep the same monthly global HCHO burden between the two versions to minimize its impact on CO fields to keep the comparison simple. We have revised it as "a scaling factor is applied to the MR results over the globe to keep the monthly $CH_2O$ production in mass consistent between the MR and LR versions".

*P6, L269: "It is noted that the MOPITT NIR/TIR retrievals, combining information from both TIR and NIR, have generally higher sensitivity to the lower troposphere compared to TCCON." Please rephrase, add a reference?*

We removed this sentence here. Their different vertical sensitivity is discussed later in the result section.

*P7 L305: "Individual aircraft profiles are assigned to certain model grid points given their geographic location and pressure levels, and all measurements are then averaged per model grid point at a 30-minute resolution to compare with corresponding model value." What would be the impact of spatial resolution? This comparison is in favor of the coarser resolution, why don't you interpolate to the observations?*

Following the reviewer's advice, we tested the interpolation of the model results to the observation's location and time. It does not impact the results.

*P7 L327: I do not understand this statement: "the total column of the model state integral is always conserved to compensate uncertainty from vertical resolution*

*change on the CTM side." Again, the evaluation at lower resolution is in favor of the coarse run.*

We rephrased the statement as "When interpolating from the vertical CTM levels to the 10 MOPITT pressure levels to obtain $\chi_{mod}$, we conserve the pressure-weighted column-mean CO concentration by a scaling factor".

*P8 L341: Typo, "of the model version or of the OH field"*
*P8 L345: change "for both OH" to "for both OH fields".*
We corrected both issues.

*P8 L350: "It is noted that although we optimize OH together with surface emissions, the system only scales slightly the big-region OH state vectors (in total six big regions over the globe) and thus the inverted surface emissions are sensitive to the prior OH fields." With OH driving the CO sinks (90 %), how would it be possible to not be sensitive to the prior OH field?*

Our point was not to overplay the effectiveness of optimizing OH, but we have deleted this sentence.

*P8 L350: "with the range showing 1-sigma standard deviation of the mean biases of all model grids, the same applies hereafter if not specified otherwise". Please remind the time period, a large variability is expected for the northern hemisphere spring.*

We revised the sentence as "with the uncertainty range showing 1-sigma standard deviation of the monthly mean biases of all model grids over three full years from 2009 to 2010".

*P8 L365: "The CO surface sources are conserved in global mass between different resolutions when emitted to the atmosphere, but with the change in resolution, the CO sink via the reaction with OH (associated with different resolutions) may differ." What is the purpose of changing the resolution if you forced to value to be equal to the coarse resolution? Those differences are what can be interested in this study, because you are comparing simulation with different spatial grid spacing.*

We keep the same surface and chemical CO sources ($CH_2O$ 3D production) in mass to reduce the features that could introduce differences in the model results. As mentioned earlier, the largest difference between the two resolutions are actually the vertical levels they have, more relevant to the features in the vertical profiles that we are interested in.

*P8 L370: "The difference between the HR and MR XCO results (~1.5 ppb) is of a much smaller magnitude than the differences between the prior and the posterior MR simulations (~10.4 ppb); it is also of a smaller magnitude than that induced by the two OH fields in the prior forward simulations when the CO sources are identical (~2.8 ppb for global average with some cancelling effect between the*

*NH and SH). The differences in modelled XCO between HR1 and HR2 are not significant at a global scale." To remove cancelling effects, could you use RMSE for comparison with measurements, and/or tropospheric abundance (tropospheric mass) for the comparison of different simulations.*

We thank the reviewer for this nice suggestion. We added information on RMSE in Fig. S1.

*P9 L388: "They also slightly overestimate XCO in SH, resulting in a smaller positive bias 2.5±3.1 ppb using INCA-OH (3.0±2.2 ppb using TransCom-OH)." Please rephrase.*

It refers to the comparison to TCCON measurements. It is rephrased as "The posterior simulations also slightly overestimate $X_{CO}$ in SH, producing a small positive bias of 2.5±3.1 ppb when using the INCA-OH field (3.0±2.2 ppb when using the TransCom-OH field) ".

*P9 L422: "when other setups being the same; here, the 1s standard deviation showing the spread across prior/posterior simulations and between the two HR models." Please do not use the average difference, is it with adjusting emissions and chemistry. You could also do a simulation with full chemistry and updated emissions.*

We agree with the reviewer that we should not average the differences between the two cases. We described their differences separately in the revised text. We did not do a full chemistry model as it is not the focus of this study. Here, we simply wanted to show the impact of a change in the vertical resolution on the CO concentration field and whether those uncertainties will change the bias structure.

*P10 L430 and Figure 3: What are the different points, is it the latitude band or the time period (Months)? A large seasonal cycle is expected (see https://www.esrl.noaa.gov/gmd/ccgg/globalview/co/co_intro.html).*

Each point represents the annual mean quantity for the surface [co] at a given station and for the $X_{CO}$ of the corresponding model grid. The structure of the biases in the seasonal cycle is not discussed in this study, as we wanted to focus on the large scale spatial features for the annual mean values.

*P10 L449 and P10 L469: title 3.4.1 MOZAIC measurements over large airports What do you mean by large airports? Did you actually select airports that are larger, in size, in number of flights?*

We did not. We revised it as "MOZAIC aircraft measurements".

*P10 L455: "are larger in the NH than in the SH, consistent with the different in prior model" Please correct to 'differences' or 'different priors'.*

We corrected it.

*P11 L511: "Such bias in representing the oceanic vertical profiles suggests error in the vertical distribution of CO source/sink over ocean or in the vertical mixing." Please rephrase, there are errors (plural) from both chemistry, horizontal and vertical transport, ocean is repeated twice.*

We revised it as "Such biases in representing the vertical CO profiles over the ocean suggest errors in the vertical distribution of secondary CO sources and its OH sinks, or in the horizontal and vertical mixing".

*P13 L600: "The choice of the prior OH seems to produce the largest differences in the simulated CO concentrations at a global scale." It is confusing, please remove 'seems', or you can say "Our study shows that the choice of. . ." Again, do not forgot that having a prescribed OH is a strong approximation (limitation), see for instance (Elshorbany et al.; 2016)*

We revised it as "the choice of the prior OH field produces the largest differences in the simulated CO concentration at a global scale across the sensitivity tests we performed in this study."

*P13 L615: "The North-to-South gradient in TransCom-OH is also closer to a recent observation-derived near equal N/S OH distribution (Patra et al., 2014)". I think those two studies are related, the TransCom-OH field is from ACTM_0.99, which is designed to have a near equal N/S OH distribution. You can mention this in the introduction, it is supposed to be the best fit to Methyl Chloroform.*

We added this information in the method section when introducing the TransCom-OH field. "The TransCom-OH field was developed by combining the semi-empirically calculated tropospheric (following Spivakovsky et al., 2000) and 2-dimensional (2-D) model simulated stratospheric distributions. It is supposed to be the best fit to Methyl Chloroform reduction rate."

*Acknowledgement Most of the acknowledgement are not respecting the recommendation, please consider contacting the instrument PI's. For instance, the MOZAIC acknowledgement (if not updated since) should be as follow:*
*The authors acknowledge the strong support of the European Commission, Airbus, and the airlines (Lufthansa, Air France, Austrian, Air Namibia, Cathay Pacific, Iberia and China Airlines so far) which have carried the MOZAIC or IAGOS equipment and un- dertaken maintenance since 1994. In its last 10 years of operation, MOZAIC has been funded by INSU-CNRS (France), Météo-France, Université Paul Sabatier (Toulouse, France) and Research Center Jülich (FZJ, Jülich, Germany). IAGOS has been additionally funded by the EU projects IAGOS-DS and IAGOS-ERI. The MOZAIC–IAGOS database is supported by AERIS (CNES and INSU-CNRS).*

We thank the reviewer for this note. We have revised the acknowledgement section carefully.

*Table 1: "List of simulations in this study", -> 'list of simulations done in this study' or 'list of simulations'.*

*Table 2: "reference" to "references"*
Corrected.

*Table 3: How are calculated the error bars (more or less)?*

We added this information into the table caption. "The uncertainty range represents 1-σ of the annual mean model-data biases across all model grids for Xco or among all stations for surface [co]."

---

## Author Comment (AC5) · 22 Mar 2018

First, we thank Dr. M. Deeter for his comments.

In general, the authors appear to understand the points I raised in my first comment, and have stated their intention to clarify these points in the revised version. I will therefore need to read the revised version before commenting further.

We have carefully revised those points. We also reduced the volume of text regarding the MOPITT algorithm following the reviewer's suggestions, since it is not our results and has been well documented.

[Figure]

However, the authors do not seem to have understood the point I raised about the V6 TIR/NIR validation results. Their response to this comment was "We cited the MOPITT papers that we consider most relevant and latest - in total four papers by Deeter et al." The issue is not whether the V6 validation paper was cited or not. The point here is that any conclusions made in the submitted manuscript regarding retrieval bias (for example, in Sections 3.2, 3.2, and 5.3) should be related to what is already known about such biases. There is valuable information in the V6 validation paper, especially regarding biases in the V6 TIR/NIR product over the Northern Hemisphere, that is completely ignored in the submitted manuscript.

We have followed this recommendation and have added more discussion regarding previous evaluation results in Section 6.2.

---

## Author Comment (AC6) · 22 Mar 2018

**Reply to short comments by H. Worden**

**Yi Yin et al.**

First of all, we thank Dr. H. Worden for her comments. And we would like to take the chance to further explain our arguments.

> **H. Worden**
>
> *The author's reply to our comment did not address the serious deficiency that their study does not provide evidence to support their conclusions about assimilating pro- file vs. column CO MOPITT retrievals, since they did not assimilate profile data, even without bias correction. It would be necessary (but still not sufficient) to show that the assimilation of profile data, without bias corrections, performs significantly worse than column assimilation as compared to independent CO observations.*

Our argument is based on existing practice (including both what we tested here and previously documented studies) and logical reasoning. We showed that (1) when updating the surface emissions, the overall shape of vertical profiles can only be marginally improved (this happens only when the profile errors stem from surface flux errors), and (2) the posterior model biases to the MOPITT profiles vary with the altitude (with opposite signs of the remaining biases between the near surface levels and the free-troposphere / stratosphere levels).

Based on these two points, one expects that when assimilating one pressure level at a time the inversion derives different estimates of surface emissions. This expectation agrees with previous experiments demonstrated by Jiang et al., (2013, 2015, 2017) assimilating the MOPITT surface level (or near surface levels), the profiles, or the total column amounts individually, and indeed, resulted in different CO emission estimates. Hence, both inverse systems show that there is some inconsistency in the vertical CO distribution between the MOPITT data and the CTM that cannot be reconciled by only updating the surface emissions.

Further extending the second point, the remaining posterior model bias to the MOPITT profile is of the opposite sign to the other independent observations we included here in this study, which is in line with a previous validation study showing that the profile retrieval has larger retrieval biases at individual retrieval level than the total column retrieval, including a more significant temporal drift (Deeter et al., 2014).

Also, when assimilating the profile, we do not have adequate information to characterize the full observation error correlations across the vertical profile, which include not only measurement errors, but also transport-model and representativeness errors, for the inversion. Taken together, it is reasonable to conclude that assimilating the total column is more robust than assimilating the profile for inverse studies.

*I disagree with their assessment that MOPITT retrieval biases cannot be corrected due to their spatial and temporal variability. Although this correction could be tedious, and it will likely be less accurate where there is a lack of validation data, bias information is provided in Deeter et al., (2014) and Buchholz et al., (2017) for MOPITT V6 and Deeter et al., (2017) for MOPITT V7.*

Our argument was made for the profile retrieval; the concern is about the error correlations and the spatial variations of the vertical structure. Taking the recent practice by Jiang et al., (2017) as an example, guided by HIPPO measurements, the authors introduced a 4-order polynomial curve to correct a latitude-dependent bias for the MOPITT profiles at each retrieval pressure level before the assimilation (as cited below, Fig. 1 from Jiang et al., 2017). This empirical approach may reduce the bias for a certain pressure level, but without any guaranty on the overall column consistency. In addition, as shown in the plot cited here, for the comparison to HIPPO measurements that are mostly sampled over the ocean where CO concentration has a relatively smooth spatial distribution, we see already a large spread in the relative biases at a certain latitude for a certain pressure level. It also remains to be further analysed whether the features over land are compatible. Bias-correction for the total column is more practical given available observations, e.g. Buchholz et al., (2017) focused on the column amounts.

[Figure]

**Figure 1.** Difference between MOPITT CO retrievals and HIPPO aircraft measurements. The aircraft measurements are smoothed with MOPITT averaging kernels. The black solid line shows the 4-order polynomial curve fitting which is used to correct MOPITT data in this work.

*After demonstrating the performance of the profile assimilation, if their conclusion about assimilating column CO vs. profiles is particular to the vertical biases in their model, they should make this more explicit in the abstract and conclusions rather than the blanket recommendation for assimilation only of column CO.*

Model errors are often highlighted in CO inverse or chemical reanalysis studies as described in the introduction of the manuscript. As stated above, differences in the vertical profile between the transport model and the MOPITT data also exist in the model used by Jiang et al. (2013, 2015, 2017), so that the inversion derives different CO emission estimates when assimilating different levels of the MOPITT profile retrievals. Gaubert et al., (2016) showed, with another model, that when assimilating the MOPITT partial profile (excluding pressure level of 300, 200, and 100 hPa) for a chemical reanalysis where the control vector being the 3D-CO field, the remaining bias in the upper troposphere (defined as less than 400 hPa) changed sign from negative in the control run to positive in the assimilation, even though the reanalysis bias in the lower troposphere remains slightly negative. The authors interpreted this bias as vertical mixing of higher CO coming from the lower troposphere being too strong. Thus, differences in the vertical profile defined by the model equation and the MOPITT data is not a specific feature that only applies to our model.

*References:*
*Deeter, M. N., Edwards, D. P., Francis, G. L., Gille, J. C., Martinez- Alonso, S., Worden, H. M., and Sweeney, C.: A Climate-scale Satellite Record for Carbon Monoxide: The MOPITT Version 7 Product, Atmos. Meas. Tech. Discuss., doi:10.5194/amt-2017-71, in review, 2017.*

*Buchholz, R. R., Deeter, M. N., Worden, H. M., Gille, J., Edwards, D. P., Hannigan, J. W., Jones, N. B., Paton-Walsh, C., Griffith, D. W. T., Smale, D., Robinson, J., Strong, K., Conway, S., Sussmann, R., Hase, F., Blumenstock, T., Mahieu, E., and Langerock, B.: Validation of MOPITT carbon monoxide using ground-based Fourier transform infrared spectrometer data from NDACC, Atmos. Meas. Tech. Discuss., doi:10.5194/amt-2016- 241, accepted, 2017.*

**References:**

Deeter, M. N., Martínez-Alonso, S., Edwards, D. P., Emmons, L. K., Gille, J. C., Worden, H. M., Sweeney, C., Pittman, J. V., Daube, B. C. and Wofsy, S. C.: The MOPITT Version 6 product: algorithm enhancements and validation, Atmos. Meas. Tech., 7(11), 3623–3632, doi:10.5194/amt-7-3623-2014, 2014.

Gaubert, B., Arellano, A. F., Barré, J., Worden, H. M., Emmons, L. K., Tilmes, S., Buchholz, R. R., Vitt, F., Raeder, K., Collins, N., Anderson, J. L., Wiedinmyer, C., Martinez Alonso, S., Edwards, D. P., Andreae, M. O., Hannigan, J. W., Petri, C., Strong, K. and Jones, N.: Towards a chemical reanalysis in a coupled chemistry-climate model: An evaluation of MOPITT CO assimilation and its impact on tropospheric composition, J. Geophys. Res. Atmos., doi:10.1002/2016JD024863, 2016.

Jiang, Z., Jones, D. B. a., Worden, H. M., Deeter, M. N., Henze, D. K., Worden, J., Bowman, K. W., Brenninkmeijer, C. a. M. and Schuck, T. J.: Impact of model errors in convective transport on CO source estimates inferred from MOPITT CO retrievals, J. Geophys. Res. Atmos., 118(4), 2073–2083, doi:10.1002/jgrd.50216, 2013.

Jiang, Z., Jones, D. B. A., Worden, H. M. and Henze, D. K.: Sensitivity of top-down CO source estimates to the modeled vertical structure in atmospheric CO, Atmos. Chem. Phys., 15(3), 1521–1537, doi:10.5194/acp-15-1521-2015, 2015.

Jiang, Z., Worden, J. R., Worden, H., Deeter, M., Jones, D. B. A., Arellano, A. F. and Henze, D. K.: A 15-year record of CO emissions constrained by MOPITT CO observations, Atmos. Chem. Phys., 17(7), 4565–4583, doi:10.5194/acp-17-4565-2017, 2017.